# CEN-tools: an integrative platform to identify the contexts of essential genes

Sumana Sharma[1,2,*,†,‡] (iD), Cansu Dincer[1,‡] (iD), Paula Weidemüller[1] (iD), Gavin J Wright[2] & Evangelia Petsalaki[1,**] (iD)

## Abstract

**An emerging theme from large-scale genetic screens that identify genes essential for cell fitness is that essentiality of a given gene is highly context-specific. Identification of such contexts could be the key to defining gene function and also to develop novel therapeutic interventions. Here, we present Context-specific Essentiality Network-tools (CEN-tools), a website and python package, in which users can interrogate the essentiality of a gene from large-scale genome-scale CRISPR screens in a number of biological contexts including tissue of origin, mutation profiles, expression levels and drug responses. We show that CEN-tools is suitable for the systematic identification of genetic dependencies and for more targeted queries. The associations between genes and a given context are represented as dependency networks (CENs), and we demonstrate the utility of these networks in elucidating novel gene functions. In addition, we integrate the dependency networks with existing protein–protein interaction networks to reveal context-dependent essential cellular pathways in cancer cells. Together, we demonstrate the applicability of CEN-tools in aiding the current efforts to define the human cellular dependency map.**

**Keywords** context-specific essentiality; CRISPR; networks; NRAS-mutant melanoma; omics integration

**Subject Categories** Cancer; Computational Biology; Methods & Resources

**Mol Syst Biol. (2020) 16: e9698**

## Introduction

A common approach to elucidate the function of a gene is to investigate the effect of its perturbation on a given biological process. Genetic perturbation experiments enable identification of genes that are essential for the survival and fitness of the cell. It is now widely accepted that the binary nature of gene essentiality as defined in classical genetics is too simplistic and gene essentiality is highly context-specific. There are genes, often termed as core fitness genes, which are required for core functioning and housekeeping of the cells and their knockout causes loss of fitness, in principle, in all cell types in all conditions. However, a large number of genes, termed context-specific essential genes, play important roles for cell fitness only in a particular genetic or environmental context (Rancati *et al*, 2017). An important field in cancer research is the identification of genes whose loss is lethal to cells only in a specific context, as the genotype-specific vulnerabilities are excellent therapeutic targets for cancer cells carrying the specific genotype without affecting normal cells.

The recent advances in gene editing technology using the CRISPR/Cas9 knockout system have enabled large-scale genome-wide screens to systematically perturb genes and rapidly identify those that are essential for proliferation and survival of cells (Shalem *et al*, 2014; Hart *et al*, 2015; Wang *et al*, 2015; Tzelepis *et al*, 2016). Initial studies on pooled essentiality CRISPR screens mainly focused on identification of therapeutically important context-specific vulnerabilities in different cancer types and mutational backgrounds (Hart *et al*, 2015; Steinhart *et al*, 2016; Tzelepis *et al*, 2016; Barbieri *et al*, 2017; Wang *et al*, 2017). In recent years, a number of studies have used the concept of co-essentiality-driven co-functionality—if essentiality profiles of two genes are correlated, the genes are likely to be involved in similar functions—to delineate novel gene functions using pooled CRISPR screens (Pan *et al*, 2018; Rauscher *et al*, 2018; Kim *et al*, 2019). Studies of this nature are possible because the number of cell lines that are being screened has increased to hundreds, and thus, it is viable to use the assumption that investigating the knockout fitness of genes across many different cell lines with different genetic backgrounds is comparable to studying different isogenic backgrounds of the same cell line. Together, the power of essentiality screens in systematic characterisation of cancer-specific genetic interaction maps of cellular processes and in the identification of therapeutically important genotype-specific vulnerabilities in cancer cells is now widely appreciated.

As the number of cell lines assayed for vulnerabilities has increased rapidly in the past few years, there has also been an

---

1   European Molecular Biology Laboratory, European Bioinformatics Institute, Wellcome Genome Campus, Cambridge, UK
2   Cell Surface Signalling Laboratory, Wellcome Sanger Institute, Cambridge, UK
    *Corresponding author. E-mail: sumana.sharma@rdm.ox.ac.uk
    **Corresponding author. E-mail: petsalaki@ebi.ac.uk
    ‡These authors contributed equally to this work
    †Present address: MRC Human Immunology Unit, John Radcliffe Hospital, University of Oxford, Oxford, UK.

increase in efforts to consolidate and standardise the essentiality screens performed in different laboratories. Recently, two major initiatives, DepMap (Meyers *et al*, 2017; Tsherniak *et al*, 2017) and Project Score (Behan *et al*, 2019), have performed essentiality screens in over 500 cell line models representing a wide range of tissue types using standard reagents and analysis pipelines. The PICKLES web server also provides a repository of all major essentiality screen initiatives, in which raw screening data are processed through a standard analysis pipeline (Lenoir *et al*, 2018). The availability of standardised essentiality screens combined with the efforts to characterise cell lines comprehensively in regard to the gene expression profiles, mutation profiles, drug responses and copy number variations from initiatives such as Cell Model Passports (van der Meer *et al*, 2019) and Cancer Cell Line Encyclopedia (CCLE) (Meyers *et al*, 2017; Ghandi *et al*, 2019) now provide a premise for data integration for systematic studies that explores the genetic vulnerabilities in a wide range of contexts.

To aid the current efforts to democratise the large-scale screens and make them accessible to a broader scientific community, we here present Context-specific Essentiality Network (CEN)-tools (http://cen-tools.com). CEN-tools is an integrated database and set of computational tools to explore context-dependent gene essentiality from pooled CRISPR data sets, obtained from the two largest publicly available essentiality screening projects: the DepMap project (Meyers *et al*, 2017; Tsherniak *et al*, 2017) and Project Score (Behan *et al*, 2019). CEN-tools offers an easily accessible web interface with built-in statistical tools, to explore statistically significant associations between the essentiality of a given gene in a user-chosen set of cell lines and a pre-defined context (e.g. mutational background, expression levels and tissues of origin). For advanced users, the python package implementation of CEN-tools also offers functions to interrogate bespoke contexts of choice to identify novel associations. We demonstrate that CEN-tools enables systematic studies to not only identify functional genetic interactions, but also to define the underlying context associated with the essentiality of a given gene. In addition, we showcase the use of CEN-tools as a new type of omics-database for integration with current omics platforms to identify biologically relevant, cancer-specific dependency networks.

# Results

## CEN-tools identifies a robust set of core essential genes

Core essential genes should, in principle, be essential in all cell types regardless of the mutational, tissue or environmental background. To avoid setting arbitrary cut-offs as to the number of cell lines that define these genes, we developed a logistic regression-based approach combined with clustering to categorise genes according to their essentiality probability profiles (Materials and Methods). The cluster of genes that exhibited high probability for being essential across all the cell lines was designated as the core essential gene cluster. We separately analysed the Project Score (SANGER) and the DepMap project (BROAD) data sets and identified 650 genes from the SANGER data set and 942 genes from the BROAD data set, assigned into the core essential gene cluster, with 519 overlapping genes between the two projects. Among these, 146 genes were previously annotated as core essential genes by the Adaptive Daisy Model (ADaM) analysis tool, which is a semi-supervised algorithm recently used to identify novel core fitness genes from essentiality screens in Project Score (Behan *et al*, 2019). We noticed that the core analysis pipeline of CEN-tools was able to capture all but 20 genes previously identified by the ADaM pipeline suggesting that CEN-tools provides a robust platform to perform core gene analysis (Fig 1A). The 20 genes from the ADaM pipeline that were not captured among the high-confidence core essential genes of CEN-tools were still identified in the SANGER data set analysis of CEN-tools. Upon closer inspection, we observed a major discrepancy in the essentiality probabilities of these 20 genes between the two projects. Genes such as *LCE1E, MED31, PISD, UBB, ALG1L,* and *HIST1H2BB* showed completely opposite essentiality profiles in the different projects (Appendix Fig S1) suggesting differences in the gRNA efficiencies for those particular genes.

To investigate how well the genes from each project were clustered, we next used the silhouette method, which compares the distance of each point to points in the same cluster with the distance to points in the neighbouring clusters. We observed that the genes that were designated into the core essential cluster in both data sets and/or also by the ADaM pipeline had higher Si-scores compared with the genes that were designated in only one of the two projects

**Figure 1. CEN-tools identifies core essential genes involved in regular housekeeping functions of a cell.**

A  Venn diagram for prediction of core essential genes by CEN-tools using both the BROAD and SANGER projects, and ADaM novel core fitness genes (Behan *et al*, 2019).

B  Box plot for Silhouette Scores (si-scores) of core essential genes predicted by CEN-tools and ADaM for the two projects BROAD and SANGER. For each project, the core essential genes were predicted separately. The centre of each box plot represents the sample median; and the ends of the box are the upper and lower quartiles; the whiskers extend to the smallest and largest observations within 1.5 times the interquartile range of the quartiles. Observations lying outside the whiskers are shown as individual data points. Box plots were drawn based on si-scores from the following numbers of genes for each project: BROAD (ADaM, SANGER, BROAD: 146, SANGER, BROAD: 373, only BROAD: 423); SANGER (ADaM, SANGER, BROAD: 146, SANGER, BROAD: 373, only SANGER: 111).

C  Pie chart for percentages of core essential genes predicted by both ADaM and CEN-tools and only CEN-tools by using both projects. The pie chart on the right panel represents the percentages of novel core genes from CEN-Tools, known core genes in pluripotent stem cells and core biological processes, and also therapeutically tractable genes annotated by the Project Score. (Behan *et al*, 2019).

D  Bar Plot of protein-complex enrichment of core genes. Y axis represents the significance of the enrichment; colours of the bars represent the percentages of the novel core genes in the complexes. The red line represents the adjusted *P*-value of 0.01.

E  Box plot for the log value of the basal expression levels of core genes in BAGEL (Hart & Moffat, 2016), ADaM, CEN-tools predictions, and non-essential BAGEL and "Not core" CEN-tools genes. The centre of each box plot represents the sample median; the ends of the box are the upper and lower quartiles; the whiskers extend to the smallest and largest observations within 1.5 times the interquartile range of the quartiles. Observations lying outside the whiskers are shown as individual data points. Box plots were drawn based on expression from the following numbers of genes for each group: Not-core : 15,330, BAGEL essential : 606, ADaM, SANGER & BROAD : 145, ADaM : 346, SANGER & BROAD : 367.

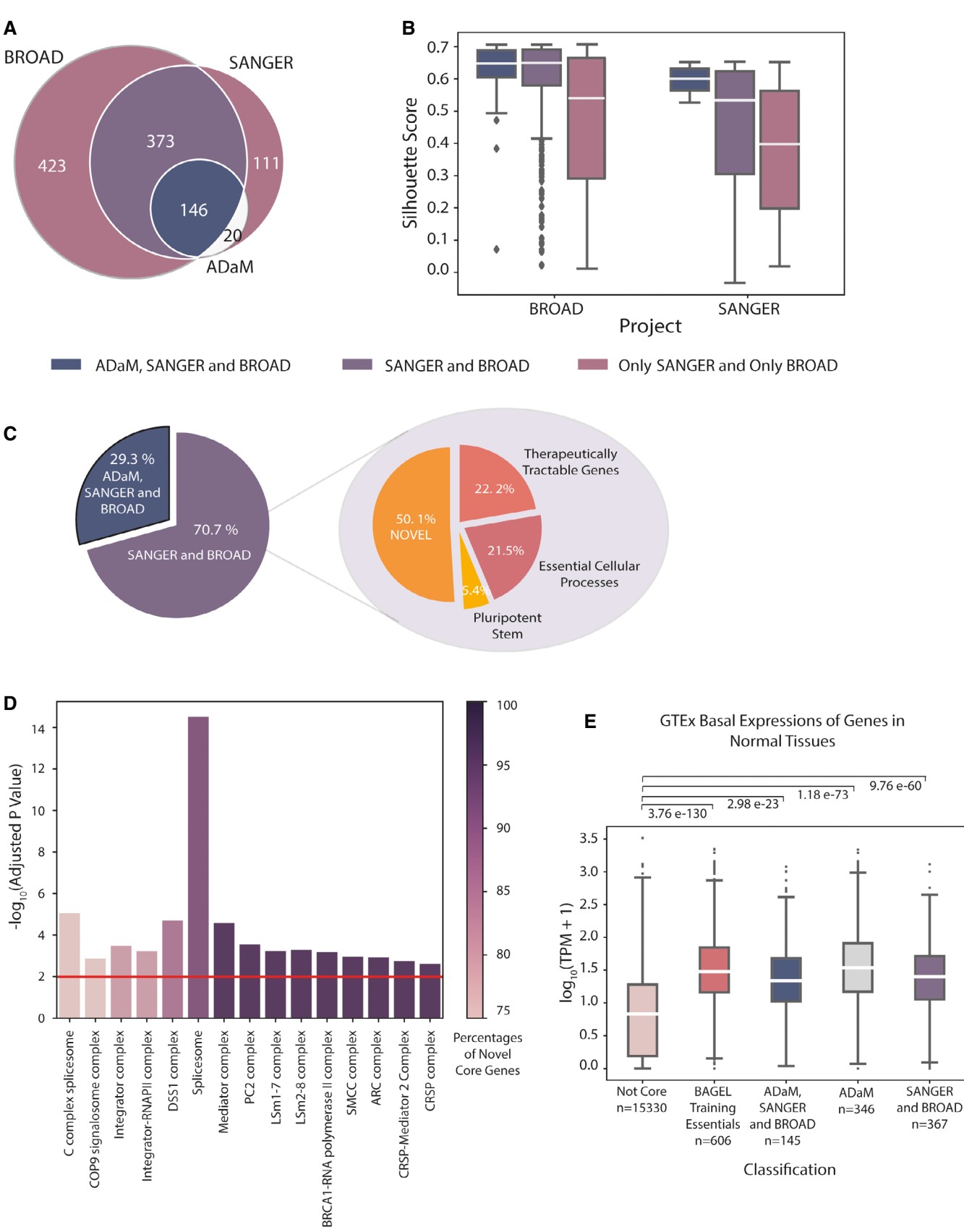

**Figure 1.**

(Fig 1B). As a higher Si-score indicates that the object is well matched to its own cluster, we defined the overlapping 519 genes from both projects to be the high-confidence set of core essential genes from CEN-tools. Of the 373 newly identified core essential genes, 80 could be assigned to known essential housekeeping complexes and processes, namely ribosomes, spliceosomes, proteasomes, DNA replication, and RNA polymerase, using the KEGG (Kanehisa *et al*, 2017) database. Another 20 genes were annotated to be essential for the fitness of human pluripotent stem cells (Ihry *et al*, 2019). To further filter genes that could be involved in fundamental cellular functions of a cancer but not a normal cell, we annotated 83 genes considered to be therapeutically tractable for multiple cancer types (Behan *et al*, 2019). This filtration step yielded a list of 190 "novel core" genes presumably important for basic housekeeping of a cell (Fig 1C).

To identify the gene families enriched in the core gene lists of CEN-tools, we first explored the previously annotated essential processes from the ADaM pipeline and were able to add new members to the pre-annotated enriched groups, such as additional members of the mediator complex *MED7, MED17* and *MED22* (complete annotation of all predicted genes in Table EV1). We also independently performed protein-complex enrichment using the CORUM database (Giurgiu *et al*, 2019), which revealed enrichment in similar complexes relating to housekeeping functions of the cells such as the DNA synthesome complex, mediator complex, integrator complex, LSm1–7 complex, and the TFIIE complex. As a new core essential complex, we also identified enrichment in the COP9 signalosome complex (Fig 1D). Since genes involved in core functioning of cells are also expressed in cells of normal tissues, we next explored the expression of the newly annotated core genes in normal human tissues from GTEx (Aguet *et al*, 2017) and observed that the basal expression of the newly annotated core genes was significantly higher than the genes that were not annotated as core essential genes (Fig 1E, Appendix Fig S2). The complete set of high-confidence core essential genes is available to download from Table EV2. The "Essentiality profile" tab of the CEN-tools web-application provides essentiality profiles for individual genes for further browsing.

## CEN-tools enables rapid interrogation of contexts to identify gene–gene relationships

The "Context Analysis" framework of CEN-tools calculates groupwise associations and correlations on a chosen set of cell lines to identify relationships between a given context and gene essentiality. To enable easy statistical comparisons, we preloaded a number of contexts on the CEN-tools website that are most likely to be used by the majority of researchers. These include tissue/cancer type-wide comparisons, essentiality correlations, correlation between essentiality and expression, essentiality driven by mutations in cancer driver genes, and correlation between drug responses and essentialities for pre-defined cell lines or user-chosen sets of cell lines (see examples in Appendix Fig S3).

Genes that are essential for a given tissue/cancer type were identified by testing if they had significantly higher essentiality compared to pancancer. Tissue-specific dependencies, however, can be driven by a number of factors such as the underlying mutation that is enriched in the given tissue type or the level of expression of

the gene in the particular tissue. Therefore, to get a better overview of different types of dependencies, we pre-calculated all possible associations for three main preloaded contexts (tissue/cancer, mutation and expression), both in pancancer and within tissue/cancer type, and represent them in Context-specific Essentiality Networks or CENs. Each edge in this network represents the type of association used to annotate the underlying contexts. To demonstrate the value of CENs, we collected the co-essentiality networks from the PICKLES database (Lenoir *et al*, 2018) and extracted the corresponding genes from our CENs for direct comparison. On the *BRAF* CEN, for example, we could identify components of the co-essentiality networks with the mutational links of *BRAF* to *MAPK1, MAP2K1, PEA15*, and *DUSP4, LIF* (Fig 2A, Appendix Fig S4). However, *SOX10, MITF* and *ZEB2* were not linked to *BRAF* itself but were instead associated with *BRAF* via their expression in the skin. *SOX10* additionally contained high-confidence self-loop edges of expression-essentiality correlation suggesting that this dependency is not directly related to *BRAF* mutational status but rather to its expression status in skin. This is consistent with the lineage specification roles it plays in skin tissue regardless of the mutational background (Harris *et al*, 2013).

In addition, the CENs are particularly useful in navigating through specific types of dependencies. For example, genes required for tissue differentiation into a particular lineage often have restricted expression, and given their cellular function, they should have high essentiality in the given tissue. Using CEN-tools, we isolated such genes (Fig 2B) and revealed a subnetwork consisting of a number of transcription factors (TFs) that are known to control tissue differentiation into a specific lineage such as *SOX10* in skin (Harris *et al*, 2013), *PAX8* in ovary, kidney and endometrium ((Grote *et al*, 2006; Cheung *et al*, 2011; Tong *et al*, 2011), and *MYCN* in neuroblastoma (Huang & Weiss, 2013). The TF *TP63* was highly expressed and essential in cell lines derived from head and neck and bladder cancers, consistent with it being a known regulator of squamous epithelium lineage (Network & The Cancer Genome Atlas Research Network, 2012). Cell lines derived from cancers of blood cells are known to have distinct lineage specification genes, and we also observed multiple specific lineage markers such as *IRF4, SPI1, GFI1, BCL2* and *MYB* (Behan *et al*, 2019) Fig 2B). *ZEB2* was associated with skin, haematopoietic and lymphoid, and soft tissue with a high statistical confidence, which is consistent with the mesenchymal origin of the cell lines from these tissue origins (De Craene & Berx, 2013). This subnetwork also revealed genes that are not necessarily lineage restricted but have an expression to essentiality relationship because of an underlying enriched mutational background. For example, the essentiality of *MDM2* in multiple tissue types was higher in cells with wild-type (WT) *TP53*. This is an observation often made in pooled CRISPR screens as cells harboring the WT tumor suppressor genes are often dependent on their repressors for proliferation (Hart *et al*, 2015). Together, these examples illustrate how networks from CEN-tools can be utilised to systematically characterise specific types of expression-related dependencies from large-scale CRISPR screens.

## CEN-tools reveals a skin-specific link between the *SOX10* transcription factor and *SRF* activity

We next examined whether tissue/cancer type-specific networks could be explored in a similar manner to identify context-specific

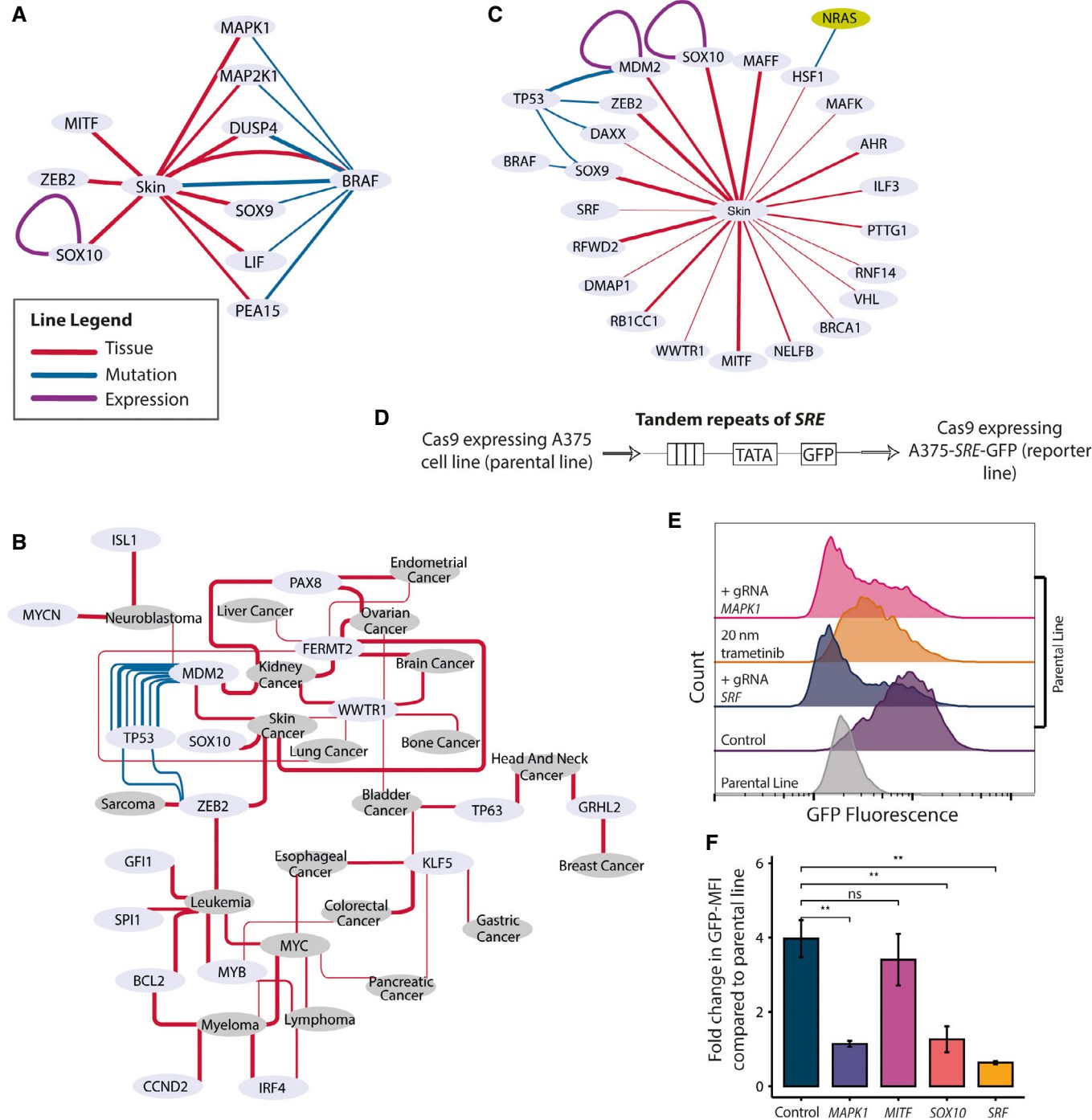

**Figure 2. Interrogation of contexts from CEN-tools identifies novel gene–gene relationships.**

A   An example of a CEN from CEN-tools. In this example, components of the *BRAF* co-essential genes from PICKLES were extracted from the CEN-tools BRAF-centric CEN network in Skin. Edges with confidence level of 2 (*P*-value < 0.01, correlation score 0.6 are depicted).

B   CENs for genes that have restricted expression and essentiality in different tissue types. The majority of the genes in this network are TFs important for lineage specification of a cell line to a particular tissue type. The width of the line in (A) and (B) denotes the confidence of association.

C   The CEN of transcription factors in skin tissue. This CEN was generated from the BROAD data set.

D   Schematic of the Cignal® lentiviral reporter construct for assessing the activity of the serum response factor (SRF) transcription factor. The construct expresses GFP under a control of a basal promoter element (TATA box) together with multiple tandem repeats of serum response element (SRE). This construct was used to generate a reporter Cas9 expressing A375 cell line for SRF activity.

E   Representative histograms depicting the GFP expression from the parental and the reporter line under different perturbations. For gRNA transductions, polyclonal lines were used to assess the GFP expression 6 days post-transduction. GFP expressions of trametinib-treated cells were measured 2 days post-treatment.

F   Bar graph depicting the GFP fold change distribution compared with that from parental line GFP distribution from three independent gRNA transductions targeting the indicated genes. Polyclonal knockout lines were used for quantification. The height of the bar graph represents the mean of fold change obtained from three replicates and the error bars depict the standard deviation. *P*-values were obtained from unpaired *t*-test; **P < 0.01; ns: not significant. Representative raw-FACS plots shrom one of the three replicates are also shown in Appendix Fig S5B.

gene function. The *BRAFV600E* mutation is a common driver mutation in melanoma cell lines and results in their addiction to the MAPK pathway. As a case study, we hypothesised that *SRF*, which is known to be activated through the MAPK pathway, would be essential in skin in the context of the *BRAFV600E* activating mutation. To test our hypothesis, we restricted our background to the skin tissue and compared the essentiality of *SRF* in the context of BRAFV600E. Surprisingly, we found no significant association between the two (Appendix Fig S5A).

To further investigate the context for *SRF* essentiality in melanoma, we extracted all TFs that were directly linked to the skin tissue, because TFs are most likely to play a central role in controlling tissue-specific gene expression. The skin TF CEN revealed a number of lineage-specific markers such as *SOX10, MITF* and *ZEB2* but also a number of other TFs whose expression is not restricted to the skin cell type (Fig 2C). We found that the essentiality of *SRF* was not associated with any enriched mutations and wondered if it is related to the expression of any skin-specific TFs.

To test our hypothesis, we focused on *MITF* and *SOX10* as the two most essential TFs in skin and used the A375 melanoma cell line harbouring the BRAF activating mutation. We generated a clonal Cas9 expressing reporter version of the A375 cell line that contained an expression cassette for GFP driven by a serum response element (SRE) promoter containing multiple binding sites for SRF (Fig 2D). We noticed that the reporter cell line constitutively expressed GFP when grown in media containing serum, which suggested that SRF was constitutively active in these cell lines. To ensure that the expression of GFP was as specific to the activity of *SRF*, we transduced the reporter cells with a single gRNA targeting *SRF*, which abolished the expression of GFP (Fig 2E). To test whether the activating mutation in *BRAF* and the consequent upstream hyperactive MAPK pathway acted on downstream *SRF* on these cell lines, we targeted components upstream of *SRF* with trametinib, which is an inhibitor of MAP2K1/2 kinase and also transduced cells with single gRNA targeting *MAPK1*. Both of these treatments led to a decrease in the GFP signal indicating that the activity of *SRF* in these cell lines was specific to the MAPK pathway (Fig 2E). While the dysregulated MAPK appeared to act directly on the activity of *SRF*, the essentiality of *SRF* in skin tissue was not related to the *BRAF* mutational status of the cells (Appendix Fig S5A). We thus tested the effect of perturbing TFs with skin restricted expression on the activity of *SRF*. While targeting *MITF* with a single gRNA did not have an effect on *SRF* activity, we noticed a significant decrease in GFP expression when *SOX10* was targeted, indicating that the activity of *SRF* was related to the expression of *SOX10* (Fig 2F, Appendix Fig S5B).

## CEN-tools uncovers essential cellular processes in cancer

Identification of mutation-dependent vulnerabilities is crucial for designing drugs that target cancer cells bearing such vulnerabilities without affecting the normal cells. To explore these vulnerabilities, we focused on the mutational associations identified in our CENs. As gain-in-function mutations in oncogenes are often associated with an increase in dependence of the cell lines harbouring the mutation, we first extracted self-loop edges connected by mutation driven essentialities in a pancancer comparison. The occurrences of oncogenic mutations, however, can be tissue-specific, so we

additionally extracted edges corresponding to mutational association between oncogenes and tissue/cancer types, which was generated by comparing the essentiality of the given cancer driver in the context of its mutation within a given tissue/cancer type. Among the most significant associations were the pancancer mutational association in genes such as *BRAF, KRAS, NRAS, HRAS, CTNNB1, PIK3CA* and *EZH2* in at least one of the two projects (Appendix Fig S6). Within Group A associations (statistical tests with six or more cell lines/group), a number of these genes were associated with tissues such as *BRAF* with Skin, *PIK3CA*-Breast and ovary, *KRAS* with pancreas, oesophagus, colon/rectum, and lung and *NRAS* with skin and hematopoietic and lymphoid, indicating the tissues in which these mutations are most relevant.

CRISPR screens performed on cancer cells with a particular genetic background are also perfectly suited to identify synthetic lethal interactions. We examined whether mutational networks of CEN-tools could be used as a guide to identify synthetic lethal interactions in tissues with a specific mutational background. As a case study, we investigated the edges corresponding to increased essentialities in NRAS mutational skin tissue background. NRAS mutations constitute 15–20% of all melanomas and are the most important sub-group of BRAF WT melanomas, yet therapeutic options for NRAS-mutant melanoma are still limited (Muñoz-Couselo *et al*, 2017). It is known that NRAS-mutant cancer cell lines rely on signalling through *CRAF* (*RAF1*) and *SHOC2* (Dumaz *et al*, 2006; Kaplan *et al*, 2012; Jones *et al*, 2019) and we could identify this association as a highly significant association in CEN-tools (Fig 3A). To identify the cellular context of these dependencies, and to highlight potential candidate target genes and pathways (Lord *et al*, 2020), we opted to integrate the dependencies from the relevant CEN with a protein–protein interaction network to identify affected cellular mechanisms. The integrated dependency-interaction network (Fig 3B) revealed a cluster of highly connected components with significant enrichment in a number of pathways including the Ras-signalling pathway, the focal adhesion pathway and the PI3K-AKT signalling pathway (Fig 3C). These represent potential target pathways for NRAS-mutant melanoma. An important protein of the Ras-signalling pathway identified within these clusters was the receptor tyrosine kinase IGF1R, for which several drugs already exist, but are not used in the context of NRAS-mutant melanoma to the best of our knowledge. A closer look revealed that the essentiality of *IGF1R* was significantly higher in *NRAS*-mutant cell lines compared with *NRAS*-WT background and also generally in *BRAF* WT melanoma cell lines, which is mainly composed of *NRAS* and *HRAS* mutants, compared with the BRAF V600E mutants (Fig 3D). In addition to *IGF1R* itself, gene encoding the FURIN protease, which is required for surface presentation of *IGF1R* (Kavran *et al*, 2014), also had a higher essentiality in *NRAS*-mutant melanoma cell lines compared cells with WT *NRAS* (Appendix Fig S7). The role of *IGF1R* in mediating acquired resistance of *BRAF* mutant melanoma cells to *BRAF* inhibitors is well-studied (Villanueva *et al*, 2010; Corcoran *et al*, 2011); however, the role of *IGF1R* in *NRAS*-mutant melanoma is not completely understood and further work will be required to establish the precise manner in which *NRAS*-mutant melanoma cell lines are dependent on *IGF1R*. Additionally, the identification of genes encoding proteins such as *ITGAV, RHOA* and *RAC1* suggest a potential role of cellular components controlling cell adhesion and cytoskeletal organisation in

NRAS-mutant melanoma. Together, this case study demonstrates that integrating the protein–protein networks with CENs provides a powerful means to identify potential synthetic lethal interactions and at the same time place the genes from dependency networks into interaction modules with enriched cellular functions for better elucidation of cell essential biological processes.

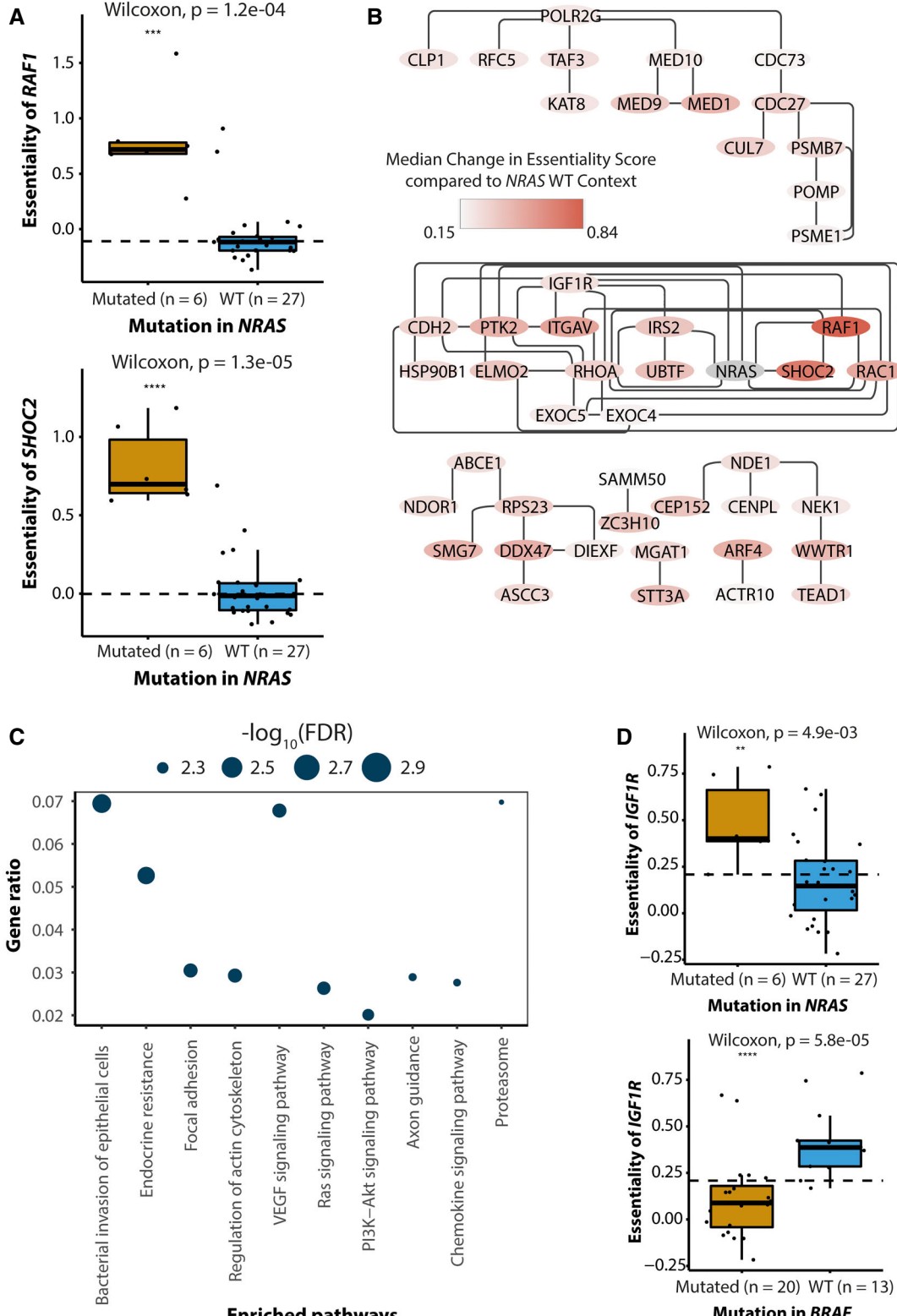

**Figure 3.**

**Figure 3.  CENs can identify cell essential processes for *NRAS*-mutant melanoma cell lines.**

A  The essentialities of *RAF1* (upper panel) and *SHOC2* (lower panel) are higher in *NRAS*-mutant melanoma cell lines (six samples) compared with *NRAS*-WT melanoma cell lines (27 samples). The centre of each box plot represents the sample median; ends of the box are the upper and lower quartiles; the whiskers extend to the smallest and largest observations within 1.5 times the interquartile range of the quartiles. Individual data points are overlayed onto the box plots. The dotted line shows the median essentiality across all cell lines displayed (regardless of group). *P*-value is obtained from a Wilcoxon test; ***$P \leq 0.001$, ****$P \leq 0.0001$.

B  Protein-protein interaction network of essential components of *NRAS*-mutant melanoma cell lines. All nodes in the network are genes whose essentiality is significantly higher in NRAS-mutant melanoma compared with *NRAS*-WT melanoma cell lines. The colour intensity represents the median-change in essential scores of *NRAS* in melanoma cell lines compared with that of *NRAS*-WT cell lines.

C  Enriched pathways in the network in (B).

D  Box plots depicting the higher essentiality of *IGF1R* in *NRAS*-mutant melanoma cell lines (6 samples) compared with non-NRAS-mutant melanoma (27 samples, upper panel) and lower essentiality of *IGF1R* in *BRAF* mutant melanoma cell lines (20 samples) compared with *BRAF* WT melanoma lines (13 samples, lower panel). The centre of each box plot represents the sample median; ends of the box are the upper and lower quartiles; the whiskers extend to the smallest and largest observations within 1.5 times the interquartile range of the quartiles. Individual data points are overlayed onto the box plots. The dotted line shows the median essentiality across all cell lines displayed (regardless of group). *P*-value is obtained from a Wilcoxon test; **$P < 0.01$, ****$P \leq 0.0001$.

## Discussion

Loss-of-function genetic screens are a powerful means to identify cellular vulnerabilities and such screens are now available for a large number of genomically and transcriptomically characterised cancer cell lines. Despite their obvious value to understanding gene functions, these screens are under-represented in omics-integrative systems biology approaches. This is largely due to the fact that the "essentiality" of a given gene is highly context-dependent, making the interpretation of essentiality from a genetic screen non-trivial and largely inaccessible to non-experts.

To address this, here we present CEN-tools, a website and accompanying python package that integrates these data. CEN-tools allows the easy navigation of the major publicly available large-scale genetic screens to identify associations between a given context and the essentiality of a gene. For this purpose, it provides a suite of statistical tools as well as integration with a number of contexts from publicly available data sets, such as gene expression, mutation background and drug sensitivity profiles of cell lines. Additionally, the "Cell Line Selection" tool within CEN-tools further enables users to restrict their analysis to their context of choice, regardless of the preloaded contexts, and make comparisons for interesting contexts such as paralog dependencies within a cell, essentialities driven by gene amplifications and essentialities associated with defects in DNA-repair mechanisms (examples in Appendix Fig S8). Further flexibility to investigate contexts of choice is also available through allowing the upload of a custom list of interesting cell line IDs. The python pipeline accompanying CEN-tools, pyCEN, also enables systematic studies using multiple genes as queries, a feature we plan to include in the future version of the website. An application for this, for example, could be to query dependencies in the context of a specific cancer subtype signature.

As a basis for CEN-tools, we identified high-confidence core essential genes, through an analysis that combined the essentiality profiles of genes from the two projects. Our core gene analysis captured almost all genes identified by ADaM (Behan *et al*, 2019), and identified 195 new core essential genes involved mainly in previously known-to-be-essential cellular processes, but also added an additional process of COP9 signalosome complex to these. The core analysis from CEN-tools could also capture all but 11 genes from the very recent core gene analysis from Dede *et al* (preprint: Dede *et al*, 2020), which used the same data sets as in this study (Appendix Fig S9). As the essentiality measured from genetic screens not only captures genes whose loss causes cell death, but

also genes whose loss results in slower proliferation of the cells, the day the experiments were performed will affect the identification of the genes essential for proliferation. This perhaps explains the higher number of core genes identified in the BROAD data sets compared with the SANGER study (21 days instead of 14 days). By directly comparing the two studies, we also identified cases in which the gRNAs' efficacy in the two projects varied considerably. Genes like *UBB*, *HIST1H2BB*, which encode for Ubiquitin B and a histone protein are very likely to be core essential but were not identified as such in the BROAD study. In some cases, it is known that "late-essential" genes are not identified as essential genes if the screen endpoint is at an earlier time-point than the time it takes for the cells with the mutation in these genes to completely drop-out of the population. However, the fact that BROAD screens had a later endpoint than SANGER screens and that these genes are known to be "early-essential" genes suggests that it is more likely that the gRNAs used for these particular genes in the BROAD study were of lower efficacy. Our high-confidence core genes were defined as those that overlapped in both projects, and hence, this core essential gene list from CEN-tools is representative of genes in which gRNA efficacy for a given gene in both studies was comparable. Users of CEN-tools can easily navigate the essentiality distributions for both projects to identify both the best timeframe and library for their specific genes of interest, when designing knockout experiments.

Using CEN-tools for group-wise testing for significant association we identified many previously described molecular markers that are associated with essentiality, including, but not limited to mutational dependence on major cancer drivers and tissue-specific lineage dependency markers (Tsherniak *et al*, 2017; Behan *et al*, 2019). We further used CEN-tools to discover and experimentally validate an association between the skin-specific gene *SOX10* and the *SRF* TF downstream of MAPK signalling in malignant melanoma. Additionally, we showed that integration of CENs with existing protein–protein interaction networks provides a powerful way to map dependencies in the context of cellular function. In the example of NRAS melanoma, in addition to identifying the expected Ras-signalling pathway, we also identified a potential role of cellular cytoskeletal processes, a pathway that is not entirely evident by only exploring individual dependencies. While these results require validations through further experiments, they demonstrate that our approach of CEN-PPI integration is both viable and novel, not yet commonly applied in the field of systems biology, to identify cellular pathway dependencies. Approaches of this nature could be refined in the

future as more context-specific PPIs (e.g. patient-specific PPI interactions) become available to ultimately aid designing better therapeutics.

It should be noted that as the current version of CEN-tools tests group-wise associations for only the given context, without taking into consideration other co-occurring contexts, there is a risk of observing confounding results if one were to only perform a single test in isolation. We thus recommend considering associations in the context of their CENs, through the network view of CEN-tools, as these will be able to point to some of the confounders, at least for the contexts whose associations have been pre-calculated. The cell line selector tool could be used for further exploration of possible confounders when interpreting associations, as it is equipped with additional information about the genetic and transcriptomic makeup of each cell line. Future versions of CEN-tools, and as more data becomes available, will integrate an analysis for confounding factors more directly. An option for this could be identifying associations between gene essentialities and a given context using a mixed effect linear model while considering defined set of contexts as covariates, an approach that has been used very recently for the identification of drug-gene associations from essentiality screens (Gonçalves *et al*, 2020). Currently, to aid the users in interpreting the statistical associations we have included a number of confidence annotations. For example, all analyses are provided at two levels of confidence: Group A analyses include more than five cell lines/group whereas Group B associations are based on the minimum three/group. Group B associations therefore should be used with caution as it is restricted by a low number of available cell lines in the data sets. All statistical associations of CENs are also annotated with low to high confidence levels depending on the *P*-value or correlation scores, enabling the users themselves to filter the associations depending on their requirements.

All systematic analysis within CEN-tools has been performed separately for the two projects. However, a recent preprint from Pacini *et al* (preprint: Pacini *et al*, 2020) has integrated the essentiality scores from SANGER and BROAD and we have included the integrated data sets in the CEN-tools website as a beta-version. We repeated our core gene analysis in this "INTEGRATED" data set (Appendix Fig S10A–D) and from a preliminary analysis, we observe that core genes identified from the core gene analysis using the integrated data set is highly concordant with the overlapping set of core genes identified from each project analysed separately. More importantly, the associations explored in detail in this study such as the essentiality of *SRF* in skin (Appendix Fig S10E) and dependence of NRAS-mutant melanoma on *IGF1R* and *FURIN* (Appendix Fig S10F and G) hold true in the integrated data set and with even higher confidence suggesting that these are indeed robust associations. Importantly, this analysis shows that increasing the number of cell lines provides increased power for detecting context-specific essential genes. As such data become available in the future and integrated in our framework, we expect CEN-tools to enable analysis in an increasing number of contexts.

In summary, we have developed a platform that can be used to explore the dependency of a given gene in a given context, which is key for elucidating the molecular function of a gene. The flexible and modular nature of CEN-tools allows for future integration with several other interesting data types, as they become available in sufficient numbers, such as drug synergy data and other context-specific information, such as protein expression, post-translational modifications and signalling signatures in specific contexts. Thus, we expect that CEN-tools will make essentiality screening data widely accessible to researchers and will facilitate its integration with orthogonal data sets, to perform systematic studies to identify cancer dependencies.

# Materials and Methods

**Reagents and Tools table**

| Resource | Resource Information | Used for | Reference |
|---|---|---|---|
| **Data sets** | | | |
| Project ScoreDependency Map (DepMap) | Genome-wide CRISPR screens (Essentiality scores- Gene depletion fold change (logFC) matrix)For BROAD, DepMap Achilles 19Q2 was used*For INTEGRATED, DepMap Achilles 19Q3 was used | Identification of Core essential geneEssentiality scores for statistical tests | Meyers *et al* (2017), Behan *et al* (2019), preprint: Dempster *et al* (2019), Broad DepMap (2019a), Broad DepMap (2019b) |
| Cancer Cell Line Encyclopedia (CCLE) | Annotations of cell lines and Mutation, CNV, expression information for BROAD and INTEGRATED projectsFor BROAD, DepMap Achilles 19Q2 was used*For INTEGRATED, DepMap Achilles 19Q3 was used | Identification of contexts | Meyers *et al* (2017), Ghandi *et al* (2019) |
| The Genotype-Tissue Expression (GTEx) | Median gene-level TPM by tissue (Analysis v8—RNASeq v.1.1.9, 2017-06-05) | Basal Level Expressions of the genes in cell lines | Aguet *et al* (2017) |
| Genomics of Drug Sensitivity in Cancer (CancerRx) | Database for drug response and therapeutic biomarkers of cancer cell lines:GDSC1: 15Oct19 version and GDSC2: 15Oct19 version | Identification of contexts | Yang *et al* (2013), Iorio *et al* (2016) |
| Cell line passports | Annotations of cell lines from SANGER project (v.20020610) Mutation information of cell lines, MSS/MSI status annotation, | Identification of contextsCell line ID mapping | van der Meer *et al* (2019) |

**Reagents and Tools table** (continued)

| Resource | Resource Information | Used for | Reference |
| --- | --- | --- | --- |
| | gene expression and CNV data sets (v.20191101) | | |
| BAGEL (Bayesian Analysis of Gene Essentiality) | Gold-standard essential and non-essential gene sets from the BAGELR implementation | TrainingBenchmarking | Hart and Moffat (2016) |
| ADaM (Adaptive daisy model) | An algorithm for identification of core fitness and context-specific essential genes in large-scale CRISPR-Cas9 screens | Comparison | Behan *et al* (2019) |
| HPSCs (Human Pluripotent Stem Cells) | Stem cell core gene set | ComparisonCore Annotation | Hart and Moffat (2016), Ihry *et al* (2019) |
| TRRUST | Human TF database | Network Annotation | Han *et al* (2018) |
| Surfaceome | Human Surface protein database | Network Annotation | Bausch-Fluck *et al* (2018) |
| SLCs (solute carrier proteins) | A group of membrane transport proteins | Network Annotation | César-Razquin *et al* (2015) |
| Kinases | Enzymes responsible for phosphorylation (important for signalling) | Network Annotation | Invergo *et al* (2020) |
| BioMart | A data mining tool for Ensembl genes, transcripts, proteins and also external information | Conversion of Ensembl IDs to official gene names | Kinsella *et al* (2011) |
| Cancer Genome Interpreter | Annotation of the genes having validated oncogenic mutations | Gene Annotation | Tamborero *et al* (2018) |
| **Software** | | | |
| dplyr (version 0.8.5) | R package for data processing | CEN-tools web server implementation | Wickham *et al* (2020) |
| ggplot2 (version 3.3.0) | R package for plotting | CEN-tools web server implementation | Wickham (2016) |
| ggpubr (version 0.2.5) | R package for plotting | CEN-tools web server implementation | Kassambara (2020) |
| gridExtra (version 2.3) | R package for plotting | CEN-tools web server implementation | Auguie (2017) |
| httr (version 1.4.1) | R package for API access | CEN-tools web server implementation | Wickham (2019a) |
| jsonlite (version 1.6.1) | R package for handling JSON data | CEN-tools web server implementation | preprint: Ooms (2014) |
| magrittr (version 1.5) | R package for data processing | CEN-tools web server implementation | Bache and Wickham (2014) |
| plotly (version 4.9.2.1) | R package for interactive plotting | CEN-tools web server implementation | Sievert (2020) |
| plyr (version 1.8.6) | R package for data processing | CEN-tools web server implementation | Wickham (2011) |
| R (version 3.6.2) | R language | CEN-tools web server implementation | R Core Team (2019) |
| rprojroot (version 1.3-2) | R package for data loading | CEN-tools web server implementation | Müller (2018) |
| shiny (version 1.4.0.2) | R package to develop shiny apps | CEN-tools web server implementation | Chang *et al* (2020) |
| shinyalert (version 1.0) | R package to develop shiny apps | CEN-tools web server implementation | Attali and Edwards (2018) |
| shinycssloaders (version 0.3) | R package to develop shiny apps | CEN-tools web server implementation | Sali and Attali (2020) |
| shinydashboard (version 0.7.1) | R package to develop shiny apps | CEN-tools web server implementation | Chang and Ribeiro (2018) |

**Reagents and Tools table**   (continued)

| Resource | Resource Information | Used for | Reference |
|---|---|---|---|
| shinyhelper (version 0.3.2) | R package to develop shiny apps | CEN-tools web server implementation | Mason-Thom (2019) |
| shinythemes (version 1.1.2) | R package to develop shiny apps | CEN-tools web server implementation | Chang (2018) |
| shinyWidgets (version 0.5.1) | R package to develop shiny apps | CEN-tools web server implementation | Perrier *et al* (2020) |
| stringr (version 1.4.0) | R package for text operations | CEN-tools web server implementation | Wickham (2019b) |
| V8 (version 3.2.1) | R package for handling javascript | CEN-tools web server implementation | Ooms (2020) |
| visNetwork (version 2.0.9) | R package for visualising networks | CEN-tools web server implementation | Almende *et al* (2019) |
| tidyr (version 0.8.3) | R package for data processing | Construction of t-sne plot for interactive cell line selector | Wickham and Henry (2019) |
| Rtsne (version 0.15) | R package for t-sne calculation | Construction of t-sne plot for interactive cell line selector | Krijthe (2015) |
| Python (version 3.6.9) | Python language | CEN-tools data curation and analysis | van Rossum and de Boer (1991) |
| NumPy | Python package for scientific computing | CEN-tools data curation and analysis | Oliphant (2015) |
| pandas | Python package for data analysis and manipulation | CEN-tools data curation and analysis | The Pandas Development Team (2020) |
| SciPy | Python package for mathematics, science, and engineering | CEN-tools data curation and analysis | Virtanen *et al* (2020b) |
| scikit-learn | Python package for predictive data analysis | CEN-tools data curation and analysis | Pedregosa *et al* (2011) |
| argparse | Python package for writing command-line interfaces | Proving user input for pyCEN | Pedregosa *et al* (2011), Davis (2019) |
| pickle | Python package for serialising and de-serialising of python objects | CEN-tools data curation and analysis | van Rossum and Team (2018) |
| matplotlib | Python package for visualisations | Data visualisation | Hunter (2007) |
| seaborn | Python package for visualisations | Data visualisation | Hunter (2007), Waskom *et al* (2017) |
| os | Python package for enabling operating system dependent functionality | CEN-tools data curation and analysis | van Rossum and Team (2018) |

## Methods and Protocols

### *Data collection and curation*

Essentiality screens for cell lines and the associated information on expression, CNV, drug response and mutation were obtained from publicly available databases (Reagents and Tools Table). In addition, a number of other data sets were used for annotation of genes and their functions (Reagents and Tools Table, direct links also available on the documentation page of CEN-tools website). Further modifications to some of the data sets were done in the following manner:

1   Gene expression values obtained from Cell Model Passports for the SANGER project were log transformed (log(FPKM + 1)).

2   Tissue annotations were obtained from both CCLE website and Cell Model Passports. In case of four cell lines (SK-PN-DW, SK-NEP-1, COG-E-352, A673), discrepancies between the two websites were found in terms of their annotations, hence their expression profiles were examined on a t-sne plot and based on these cells were manually annotated as "Bone" tissue. In case of discrepancies in cancer type annotations, the Cellosaurus (Bairoch, 2018) under the Swiss Institute of Bioinformatics was further consulted and the manual annotations were provided. The complete tissue/cancer type annotations are provided in Table EV3.

3   Corrected log fold changes from Project Score and DepMap projects were retrieved from Project Score, in which the same preprocessing pipeline was applied. "Essentiality scores" were calculated by computing the scaled log2 fold change. For this, gene-level fold changes were first quantile normalised per sample and then median scaled according to BAGEL essential and non-essential genes (Hart & Moffat, 2016) as applied in (Gonçalves *et al*, 2020). The scaled values were then multiplied by $-1$ such that essential genes have a median $\log_2$ fold change of 1 and non-essential genes a median $\log_2$ fold change of 0.

4   To make the comparison between the two projects feasible, all systematic analysis on the SANGER and BROAD individual data sets was restricted to the 16,819 genes targeted in both projects.

### Identification of core essential genes

To identify core genes that are essential for all cell types, we used the corrected log-fold-change (logFC) matrix calculated using the MAGeCK (Li *et al*, 2014) pipeline post CRISPRcleanR (Iorio *et al*, 2018) correction from the Project Score analysis pipeline. Rather than using strict cut-offs to identify essential genes from each data set, we opted to convert the logFC values into essentiality probabilities and compare the probability of each gene being essential from the two independent data sets to identify core essential genes. To this end, we applied the following three steps (Appendix Fig S11A):

1   Training a logistic regression function that can separate genes as essential and non-essential inside each cell line.
    We separately trained logistic regression models with gold-standard reference BAGEL essential and non-essential gene lists from BAGEL (Hart & Moffat, 2016, also see Table EV1 for the genes used for training in each study). For each project, we separately applied these models on the remaining genes and assigned probabilities of each gene being essential in each cell line (ROC and PR Curves in Appendix Fig S11B).

2   Extracting probability distributions of the predictions for being essential of each gene across cell lines and finding the patterns of the distributions.
    To calculate the patterns of probabilities for a gene being essential across cell lines, we first sorted the essentiality probabilities in an increasing order and converted the continuous probability distribution into 20 bins of discrete frequency of probabilities. This generated a numeric matrix of probability frequencies for each gene being essential.

3   Clustering the essentiality patterns using k-means and defining core essential and non-essential gene clusters as described in Appendix Fig S11A and shown in Appendix Fig S11C.
    We used these matrices as input for unsupervised *k*-means clustering. Using the silhouette method, we identified the optimum number of clusters to be 4. The silhouette method compares the distance of each point to points in the same cluster with the distance to points in the neighbouring clusters. A high silhouette score means that the points are well placed within a cluster and the clusters are well separated, allowing a robust definition of the number of clusters and the assignment of samples.

The four major clusters of probabilities represented the different profiles of gene essentialities as shown in Appendix Fig S11C. We identified Cluster 1 as the cluster that best represented core essential genes as the genes in this cluster had probability distribution skewed to 1. Cluster 4 genes conversely showed the probability distributions skewed to 0 and were therefore labelled as non-essential. The remaining two clusters indicated context-specific (Cluster 2) and rare-context-specific (Cluster 3) essential genes as their probability distribution revealed essentiality fluctuations across the cell types (Table EV1).

### Contexts interrogated in CEN-tools and representation of the dependency networks (CEN)

The pre-defined contexts in CEN-tools were split into either "discrete" or "continuous context" and were broadly classified into four major groups:

1   Tissue/cancer (discrete context): Tissue refers to the tissue of origin of the cell lines. Cancer refers to the further classification of the cell lines into the cancer type within each tissue. Table EV3 shows the different tissues and cancer types available in the two different databases and the relationship between the cancer types with the tissue of origin. For statistical testing, the chosen tissue was used as the "test" group.

2   Mutation (discrete context): For mutation analysis, two different mutational annotations obtained from CCLE database (Meyers *et al*, 2017; Ghandi *et al*, 2019) and Cancer Genome Interpreter (Tamborero *et al*, 2018) were used. In all two-group statistical tests, for a given gene, the cell lines with the mutated gene were used as the "test" group and those with non-mutated genes were used as the "control" group.

   a   CCLE—Hotspot mutation: Pre-annotated commonly occurring hotspot mutations in 75 genes were used in the analysis Specific protein changes for the hotspot mutations used in this study are detailed in Table EV4.

   b   Cancer Genome Interpreter Oncogenic mutation: A less stringent mutational analysis was performed by using *any* mutation in an oncogene A total of 76 oncogenes were annotated (Table EV4).

3   Expression (continuous context): For expression analysis, cell line expression values were obtained from the Cell Line Passport and CCLE database as normalised FPKM values (Meyers *et al*, 2017; Ghandi *et al*, 2019; van der Meer *et al*, 2019) for SANGER and BROAD, respectively.

4   Drug response (continuous context): Drug responses for the cell lines were obtained as normalised Z-scores from the CancerRxGene database (Yang *et al*, 2013; Iorio *et al*, 2016). These are only available for a subset of cell lines and are not included in the pre-calculated CEN contexts but can be navigated through the Context Analysis tab.

Contexts were first separated into continuous or discrete type contexts. For discrete contexts, cell lines were separated into test and control groups depending on whether they fulfilled the criteria of the context or not. The groups were separated either from pancancer or within a specific tissue/cancer type. We opted for non-parametric tests for group-wise testing mainly because the scaled essentiality scores were not normally distributed and also in some cases the number of samples in each group was fewer than 20. For the statistical tests, Kruskal–Wallis and Mann–Whitney *U* (two-samples Wilcoxon) tests were used with default parameters (*SciPy* v.1.4.1 package (Virtanen *et al*, 2020a) in Python v. 3.7.4 and *stats* and *ggpubr* package (Kassambara) in R (R Core Team, 2019)). For all statistical tests, the number of samples in each group was at least 3. For further confidence annotation, associations were divided into Group A type associations, in which there were at least six samples per group, and Group B associations, in which there were only 3–5 samples per comparison group.

For continuous contexts, Pearson correlation was used (*SciPy* v.1.4.1 package (Virtanen *et al*, 2020a) in Python v. 3.7.4 and *stats*

package in R (R Core Team, 2019)). Correlation calculations are provided between the selected group and pancancer as well as within the relevant tissue/cancer type. For the generation of CENs, we required at least five samples to perform correlation tests.

The statistical tests yielded a large number of associations between genes with the pre-defined contexts. To extract meaningful relationships, we iteratively searched pre-defined contexts in all genes in a given project and used a network visualisation approach to represent all statistical relationships in CENs.

### Extraction of BRAF co-essential CEN (Fig 2A)

The BRAF co-essential CEN was extracted by subsetting the Network file (downloadable from downloads page of CEN-tools) to only include the genes represented in the PICKLES database as BRAF co-essential genes ('BRAF', 'MAPK1', 'MAP2K1', 'PEA15', 'SOX10', 'SOX9', 'MITF', 'DUSP4', 'ELOA', 'ZEB2', 'NFATC2', 'LIF'). Further subsetting was then performed on the network file as follows:

- Project = 'BROAD'
- comparision_in = 'Skin'
- effector = 'expression' or effector = 'tissue' or effector = 'hotspot_mutation' or effector = 'hotspot_co_mutation'.
- higher = 'high_in_group' or is.na(higher)
- median_group > 0 | is.na(median_group)
- width ≥ 2
- dashes = FALSE

The resulting subnetwork was imported in Cytoscape (Shannon *et al*, 2003) to visualise the network. Note that the PICKLES co-essential genes are slightly different than the CEN-tools top10 co-essential genes, due to the use of different data sets in the two projects.

### Extraction of lineage CENs (Fig 2B)

Lineage-specific CENs were extracted by subsetting the BROAD project using "Expression" as the effector. The entire network was then subsetted to contain only the interactions of this gene set. To download this network and the node attributes from the CEN-tools website, the following steps were taken:

1    Navigate to the "Network analysis" tab.
2    Choose the following parameters:
  a   Basic parameters:
    i   Project: BROAD
    ii   Generate CEN centred around: Cancer type
    iii   Tissue of Origin: All
    iv   Toggle "Expression-specific"
    v   Display effector edges corresponding to: Select all (Mutation, Expression, Tissue/Cancer)
  b   Advanced edge filter options
    i   For Expression context correlations show edges corresponding to: Positive correlations
    ii   For Tissue/Cancer context comparisons show edges corresponding to: Increase in essentiality/expression
    iii   For Mutation context comparisons show edges corresponding to: Both Increase and decrease in essentiality/expression
    iv   Select mutation annotations: CCLE: hotspot mutations

    v   Only show tissue/cancer edges in which the median essentiality score of the essential context is higher than: 0.3
    vi   Only show mutation edges in which the median essentiality score of the essential context is higher than: 0.4
    vii   Perform context comparisons: Within a tissue of origin/cancer type.
    viii   Confidence level of association (Tissue/Cancer type): 1: Low
    ix   Confidence level of association (Mutation): 2:Medium
    x   Select: Group A
3    Press "Initialise network"
4    Download the network and the node attributes by clicking the download buttons in the left sidebar. Open the network file in Cytoscape (Shannon *et al*, 2003).

### Extraction of skin-specific CENs (Fig 2C)

Skin-specific CENs were extracted from the "BROAD" project. To download this network and the node attributes from the CEN-tools website the following steps were taken:

5    Navigate to the "Network analysis" tab.
6    Choose the following parameters:
  a   Basic parameters:
    i   Project: BROAD
    ii   Generate CEN centred around: Tissue
    iii   Tissue of Origin: Skin
    iv   Display effector edges corresponding to: Select all (Mutation, Expression, Tissue/Cancer)
  b   Advanced edge filter options
    i   For Expression context correlations show edges corresponding to: Positive correlations
    ii   For Tissue/Cancer context comparisons show edges corresponding to: Increase in essentiality/expression
    iii   For Mutation context comparisons show edges corresponding to: Increase in essentiality/expression
    iv   Select mutation annotations: CCLE: hotspot mutations
    v   Only show tissue/cancer edges in which the median essentiality score of the essential context is higher than: 0.2
    vi   Only show mutation edges in which the median essentiality score of the essential context is higher than: 0.4
    vii   Perform context comparisons: Within a tissue of origin/cancer type.
    viii   Confidence level of association (Tissue/Cancer type): 1: Low
    ix   Confidence level of association (Mutation): 2: Medium
    x   Select: Group A
7    Press "Initialise network". The network will have too many nodes to be displayed but will still be generated (you will see a warning).
8    Download the network and the node attributes by clicking the download buttons in the left sidebar. Open the network file in Cytoscape.
9    For visualisation, only display nodes with "TF" attribute and any other nodes directly associated with these "TF" nodes.

### Integration of CEN with human interactome (Fig 3B)

To elucidate the cellular essential processes of the NRAS-mutant skin cancer cells, we integrated the NRAS-mutant skin CEN,

comprising the statistical associations between the gene dependencies, with the STRING protein interaction network, comprising biological associations in the human cells (Szklarczyk *et al*, 2019). This analysis was performed using the BROAD data set on the CEN-tools website using the following steps.

1 Navigate to the "Network analysis" tab.
2 Choose the following parameters:
 a Basic parameters:
 i Project: BROAD
 ii Generate CEN centred around: Gene
 iii Selected query gene: NRAS
 iv Tissue of Origin: Skin
 v Display effector edges corresponding to: Select all (Mutation, Expression, Tissue/Cancer)
 vi Toggle below to display edges corresponding to either Tissue or Cancer subtype: Tissue
 b Advanced edge filter options
 i For Expression context correlations show edges corresponding to: Positive correlations
 ii For Tissue/Cancer context comparisons show edges corresponding to: Increase in essentiality/expression
 iii For Mutation context comparisons show edges corresponding to: Increase in essentiality/expression
 iv Select mutation annotations: CCLE: hotspot mutations
 v Only show tissue/cancer edges in which the median essentiality score of the essential context is higher than: 0.2
 vi Only show mutation edges in which the median essentiality score of the essential context is higher than: 0.4
 vii Perform context comparisons: Within a tissue of origin/cancer type.
 viii Confidence level of association (Tissue/Cancer type): 1: Low
 ix Confidence level of association (Mutation): 1: Low
 x Select: Group A
3 Press "Initialise network". A network will appear.
4 On the main panel, press the toggle button "Filter 1: Hide edges corresponding to change only in expression". This will update the network
5 On the side panel, press the "Load tissues represented in the current network".
6 Upon clicking the button in step 5 a dropdown menu with "Filter to only show edges corresponding to following tissues (press 'refresh' to update the list if filtering parameters are changed)" will appear. Select "Skin" on from this dropdown menu.
7 Download the networks and the node attribute files for the shown networks. These files will be used later for network annotation.
8 Press "Map the current nodes to a PPI network" button. This will automatically trigger switching to a new tab "Integration with PPI network".
9 On the Integration with PPI network tab, select the following options:
 a Display Network from STRING: Map CEN onto STRING network.
 b Select both output options:
 i Show CEN-mapped STRING network: Upon clicking this button a purple button "Retrieve PPI network of CEN from STRING" will appear. Press this button to visualise the network on the main panel. Use, adjust STRING interaction score: 400.
 ii Perform enrichment analysis and further choose the following options.
 1 Choose the enrichment category to display: KEGG
 2 Choose enrichments FDR cut-off: 0.05
10 Download the network and the node attributes by clicking the download buttons in the left sidebar. Open the network file in Cytoscape (Shannon *et al*, 2003).

### Cell line selection tool

To allow users to choose specific cell lines to explore using CEN-tools, our Shiny app implementation offers the option to upload of a list of cell lines. For a more explorative and interactive approach, the user has also the option to custom select cell lines using our "Interactive Cell Line Selector". Using simple dropdown menus, the user can restrict the choice of cell lines based on mutations, copy number variations (CNV), growth property, microsatellite stability, tissue and cancer subtype. Only cell lines for which we had essentiality data, mutation information and expression data are available for selection.

Selected cell lines are presented in a table and highlighted in a T-distributed Stochastic Neighbour Embedding (t-SNE) plot for further interactive selection. In the t-SNE plot, each cell line is represented by a dot. Cell lines with similar gene expression profiles will be closer together. The t-SNE plot was generated with the following steps:

1 The gene expression matrix of each project (BROAD, SANGER and INTEGRATED) containing FPKM values with genes as columns and cell lines as rows was reduced to two dimensions using the *Rtsne* package (version: 0.15; jkrijthe) in R (version: 3.6.1). Default parameters were used except for perplexity, which was set to 70 and the number of maximal iterations, which was changed to 500.
2 The t-SNE plot was generated with the R *ggplot2* package using the scatterplot function *geom_point* (version: 3.3.0) (Wickham, 2019a,b) with the two t-SNE dimensions as x and y axes, respectively. Cell lines with similar expression profiles are close together in the two t-SNE dimensions.
3 The user can select cell lines interactively by either clicking on the dots of interest or by drawing a window over a number of cell lines and hitting the "Select points" (only select the chosen points) or "Add points to selection" (add the choice to previously selected cell lines) button. More information on the selected cell lines is shown in the corresponding table.
4 To support the decision, the dots can be highlighted by tissue of origin or subtype.
5 The choice of selectable dots can be restricted by using the above-described dropdown menus for mutations, CNV's etc. on the left panel of the app. Cell lines that do not adhere to the chosen criteria will appear as faint grey dots in the t-SNE plot and cannot be clicked. Cell lines that adhere to the chosen criteria but are not part of the selection appear as grey dots and can be clicked to add them to the selection.
6 As soon as the user is satisfied, the selected cell lines can be submitted for further Context analysis to CEN-tools by hitting the "Click here to confirm your selection" button. The user

should also give their selection a name that will be later on used as a label in plots of subsequent analysis.

Advanced python tool users of CEN-tools are also able to restrict the analysis to their choice of cell lines. For detailed guidance on the python tool refer to the CEN-tool python documentation.

### Investigating paralog dependency (Appendix Fig S8A)

The association of *RPL22* mutation and the essentiality of its paralog *RPL22L1* was investigated in the "BROAD" project. The following steps were taken:

1  Navigate to the "Context analysis" tab and "Tissue/Coessentiality" subtab.
2  On the left menubar select "Advanced selection" and then "Interactive Cell Line Selector"
3  Hit the "Launch the interactive Cell Line Selector" button. You will be redirected to a new tab. Wait a little while until the interface is fully loaded.
4  On the left menubar choose:
   a  Show cell lines with mutations from: Input genes
      i  Type "RPL22" in the appearing text box
      ii  Hit the "Submit" button
      iii  Should cell lines contain a mutation in all ('AND') or at least one ('OR') of the above chosen genes?: OR
   b  Selection based on Copy Number Variation (CNV)?: No selection based on CNV
   c  Cell culture growth properties: All
   d  Genome stability: All
   e  Tissue of origin (multiple selection allowed): All
   f  Cancer type (multiple selection allowed): All
   g  Colour cell lines by: Tissue
   h  Choose a name for your selection: e.g. "RPL22 non-silent mutation"
5  Hit the "Click here to confirm your selection" button. You will be redirected to the "Context analysis" tab
6  On the left menubar select:
   a  Start typing to select gene: RPL22L1
   b  Cells not chosen will be labelled as "Pancancer", do you wish to subset Pancancer list by tissue type?: No

### Investigating essentiality based on CNV status (Appendix Fig S8B)

The association of *ERBB2* amplification with its essentiality in breast and oesophagus cell lines was investigated in the "BROAD" project. The following steps were taken:

1  Navigate to the "Context analysis" tab and "Tissue/Coessentiality" subtab.
2  On the left menubar select "Advanced selection" and then "Interactive Cell Line Selector"
3  Hit the "Launch the interactive Cell Line Selector" button. You will be redirected to a new tab. Wait a little while until the interface is fully loaded.
4  On the left menubar choose:
   a  Show cell lines with mutations from: No selection based on mutation
   b  Selection based on Copy Number Variation (CNV)?: Yes
      i  Wait until you see a black box below the slider. This might take a while.
      ii  Choose a range of relative copy numbers to restrict the choice of genes with CNV: 1.1-7.22

   iii  Pick 1 or more genes, whose relative CN lies within the selected range: ERBB2
   iv  Should cell lines contain a CNV in all ('AND') or at least one ('OR') of the above chosen genes?: OR
   c  Cell culture growth properties: All
   d  Genome stability: All
   e  Tissue of origin (multiple selection allowed): Breast, Oesophagus
   f  Cancer type (multiple selection allowed): All
   g  Colour cell lines by: Tissue
   h  Choose a name for your selection: e.g. "ERBB2 amplification"
5  Hit the "Click here to confirm your selection" button. You will be redirected to the "Context analysis" tab.
6  On the left menubar select:
   a  Start typing to select gene: ERBB2
   b  Cells not chosen will be labelled as "Pancancer", do you wish to subset Pancancer list by tissue type?: Yes
   c  Subset Pancancer to a specific tissue of origin: Breast, Oesophagus

### Investigating essentiality based on microsatellite instability status (MSI) (Appendix Fig S8C)

The association of microsatellite instability (MSI) with the essentiality of the *WRN* helicase in colorectal cell lines was investigated in the "SANGER" project. The following steps were taken:

1  Navigate to the "Context analysis" tab and "Tissue/Coessentiality" subtab.
2  On the left menubar select "Advanced selection" and then "Interactive Cell Line Selector".
3  Hit the "Launch the interactive Cell Line Selector" button. You will be redirected to a new tab. Wait a little while until the interface is fully loaded.
4  On the left menubar choose:
   a  Show cell lines with mutations from: No selection based on mutation
   b  Selection based on Copy Number Variation (CNV)?: No selection based on CNV
   c  Cell culture growth properties: All
   d  Genome stability: MSI
   e  Tissue of origin (multiple selection allowed): Colon/Rectum
   f  Cancer type (multiple selection allowed): All
   g  Colour cell lines by: Tissue
   h  Choose a name for your selection: e.g. "MSI in colorectal"
5  Hit the "Click here to confirm your selection" button. You will be redirected to the "Context analysis" tab.
6  On the left menubar select:
   a  Start typing to select gene: WRN
   b  Cells not chosen will be labelled as "Pancancer", do you wish to subset Pancancer list by tissue type?: Yes
   c  Subset Pancancer to a specific tissue of origin: Colon/Rectum

### Applying advanced thresholds for network generation in CEN-tools

The network tab of CEN-tools offers advanced options to users, which can be used to apply a range of thresholds to identify the most CENs for their study. In this study, we have used a multiple thresholding system depending on how many confounding factors one would expect within the group being tested and the purpose of the study as detailed below:

1  Lenient threshold (median essentiality score of the essential context is higher than: 0–0.2, confidence level of association: low) is generally applied for "Tissue" context in which there is likelihood of a large number of confounding factors (e.g. different mutational backgrounds and cancer subtypes), therefore a higher dynamic range of essentiality score distributions. In some cases, tissue contexts will have a very high dynamic range and median essentiality score can be less than 0.2 even though the association is of high statistical confidence ($P < 0.01$). It is possible to add these edges; however, users should avoid adding confidence 1 edges ($P$-values 0.01–0.05) for tissues that also have low median score to avoid including high number of false-positive edges.

2  Medium threshold (median essentiality score of the essential context is higher than: 0.3, confidence level of association: low) is generally applied for "Cancer type" contexts (which would still have stratifications according to mutational backgrounds, but a smaller dynamic range of essentiality score distributions).

3  Stringent threshold (median essentiality score of the essential context is higher than: 0.4, confidence level of association: medium) is applied when the tested context is a single mutation in a single gene in a single tissue/cancer type.

4  CENs that are used in consort with PPI integration are generated with a medium threshold to identify all possible enriched complexes.

Users should note that these thresholds are recommendations and they may choose to use a different thresholding method depending on the requirement.

### Experimental methods

#### A375 cell culture

The Cas9-expressing A375 cell line was obtained from the Sanger Institute Cancer Cell Line Panel (https://cancer.sanger.ac.uk/cell_lines). The cell line was cultured in DMEM/F12 media (Life Technologies) supplemented with 10% heat-inactivated (50°C for 20 min) FBS, 20 µg/ml blasticidin and penicillin–streptomycin at 37°C with 5% $CO_2$. Logarithmic growth phase of the cells was maintained by passing the cells every 2–3 days. Cells were tested and found to be mycoplasma free.

#### Lentiviral production and transductions

All lentiviruses were produced, and all transductions were performed as described before (Sharma & Wright, 2020). All gene-specific gRNAs were obtained from Sanger Whole Genome CRISPR Arrayed Libraries (Metzakopian *et al*, 2017), and lentiviruses were produced in HEK293-FT cells using the Addgene lentiviral packaging mix. Polybrene (8 µg/ml) was added to the cells prior to the transduction of A375 cells using the "spinoculation" procedure.

#### Generation of the reporter A375 cell line

The reporter line of A375 was generated by transducing Cas9 expressing A375 cell line with a commercial reporter construct for SRE activity (Cignal Lenti SRE Reporter (GFP) Kit: CLS-010G). To establish a clonal reporter line, 8 days post-transduction, cells were individually sorted into 96-well plates using FACS (MoFlo XDP). The clonal cells were further cultured for 3 weeks, and 48 individual clones were tested for their expression of GFP. Five clones exhibiting different levels of GFP were selected for expansion. The selected clone exhibited the highest constitutive expression of GFP when grown in culture

supplemented with 10% serum and the highest loss of GFP signal when transduced with a single gRNA targeting *SRF*.

#### Flow cytometry

Drug treatments and gRNA transductions were performed on six-well culture plates with $1 \times 10^6$ cells/well. Treated cells were detached from culture plates using EDTA, washed twice with PBS supplemented with divalent ions and analysed using flow cytometry. All flow cytometry was performed on a Cytoflex flow cytometer. 10,000 events per sample was acquired for each sample. Live cells were gated using forward and side scatter. GFP was excited at a wavelength of 488 nm and emission detected using a 530/30 band pass filter; BFP was excited at 405 nm and the emission detected using a 450/50 band pass filter. Analysis was performed using FlowJo software (Treestar).

**Expanded View** for this article is available online.

### Acknowledgements

SS, CD, PW and EP and the manuscript publication costs are funded by EMBL-EBI core funding. GJW and experiments were funded by Wellcome Trust (grant 206194). We would like to thank the Wellcome Sanger Institute flow cytometry core facility for help with flow cytometry experiments; James Klatzow for help with core gene analysis; Ijaz Ahmad for help with setting up the web server and the members of the Petsalaki group, Toby Gurran, Enrica Bianchi and Mathew Garnett for critical reading of the manuscript and testing of the CEN-tools web server.

### Author contributions

Conceptualisation: SS, EP; Data curation, methodology, formal analysis, investigation, validation: SS, CD; Visualisation, software: SS, PW; Software: CD; Writing—original draft: SS, CD, PW; Writing—review & editing: EP, GJW. Supervision: SS, EP, GJW; Resources and funding acquisition: GJW, EP; Project administration: SS, EP.

### Conflict of interest

The authors declare that they have no conflict of interest.

## Data availability

CEN-tools is available as a user-friendly web server under http://cen-tools.com. The data sets and computer code produced in this study are available as follows:

- Analysed data/Results: Biostudies (Sarkans *et al*, 2017): S-BSST479 (https://www.ebi.ac.uk/biostudies/studies/S-BSST479)
- Code:
  ○ Python package: GitLab (https://gitlab.ebi.ac.uk/petsalaki lab/centools_pycen)
  ○ Other analyses: GitLab (https://gitlab.ebi.ac.uk/petsalakilab/ce ntools)
- The web server also includes all data in the downloads tab.
- The Docker image of the web server is available here: ftp://ftp.eb i.ac.uk/pub/contrib/petsalaki/CEN-tools/. A detailed guide on using the docker image is provided in the appendix section that is also in the same link.

All code and data are open source and freely available under a GPL license.

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
