## [Review Process File · Molecular Systems Biology]

CEN-tools: An integrative platform to identify the 'contexts' of essential genes.

Sumana Sharma, Cansu Dincer, Paula Weidemüller, Gavin Wright, and Evangelia Petsalaki
DOI: [10.15252/msb.20209698](https://doi.org/10.15252/msb.20209698)

Corresponding author(s): *Evangelia Petsalaki* (petsalaki@ebi.ac.uk), *Sumana Sharma* (sumana@ebi.ac.uk)

Review Timeline:	Submission Date:	9th May 20
	Editorial Decision:	10th Jun 20
	Revision Received:	14th Aug 20
	Editorial Decision:	9th Sep 20
	Revision Received:	15th Sep 20
	Accepted:	21st Sep 20

Editor: Maria Polychronidou

Transaction Report:

Thank you again for submitting your work to Molecular Systems Biology. We have now heard back from the three referees who agreed to evaluate your study. Overall, the reviewers think that CEN-tools seems potentially useful. They raise however a series of concerns, which we would ask you to address in a revision.

As you will see below, the reviewers make constructive suggestions on how to improve the study. I think that their recommendations are rather clear and I see no need to repeat all the points listed below. I would only like to mention two specific points:

- As you will see, reviewer #2 is concerned about the novelty of CEN-tools. We would ask you to clarify the novelty of CEN-tools compared to the DepMap functions mentioned by Reviewer #2.
- Reviewer #1 points out that the web interface needs to be as user-friendly as possible, to make sure it is widely used and easily accessible to non-bioinformaticians. Getting feedback on this from wet lab scientists might be helpful.

Reviewer #1:

The authors present a tool to navigate gene essentiality in different cellular contexts using large-scale CRISPR-Cas9 loss of function screens in human cell lines. Such large scale datasets are difficult to navigate for non-specialists but are a rich source of data to understand the relationships among genes. To fully exploit these resources, we need tools that allow to extract information among the importance of genes across cell lines, the similarity among genes, etc. Here, Sharma et al present such a tool that integrates information on some of the largest CRISPR screens that have been performed thus far.

Major comments:

I believe integration webtools such as CEN-tools.com are very useful for a quick look up for one's preferred gene. People who are at ease with the analysis of such data may not need the web interface. I would therefore encourage the authors to make the web interface as user friendly as possible. For instance, if I go to Network Analysis, I am not sure what I am looking at in terms of network. It would be useful to have a description of what each page can do at the top, giving instructions to the user and for instance say: if you select a tissue type and X, CEN will show you a network that represents XX and YY. If the tool is not easy to use for non-bioinformaticians, it may be of limited use because bioinformaticians may not need it.

As reported by the authors, they have identified a set of core-essential genes that are robustly essential (context-independent). These genes are contained in one of few clusters they identify based on the essentiality probability distribution (Cluster #1 in Figure EV10 panel C). The genes in this cluster distinctively show a high probability of being essential. However, this cluster only classifies ~3-5% of the genes. Among the rest of the genes (97-95%) classified in other clusters, I wonder if it would be possible to identify genes that are robustly non-essential and those that commonly show context dependent essentiality. Such information, if it is possible to obtain, may add an additional utility to the CEN-tools method.

From the essentiality probability distributions (as shown in Figure EV10 panel C), it appears that a fraction of genes show robustly unimodal essentiality probability distributions (Cluster #3 and #4) with their peaks at low probability for being essential. Could such genes be classified as robustly non-essential?

Overall, what fraction of the genes show intermediate probability of being essential, representing genes that commonly show a context dependent essentiality?

Minor comments:

1. It would be useful to include page numbers for peer review
2. For the sake of stringency, in the integration of CEN with human interactome, authors excluded "dependent genes having median essentiality below than or equal to -1.5 in NRAS mutant skin cell lines, edges having STRING interaction score below 400". In my opinion, the authors should explicitly mention the rationale for setting such thresholds.
3. Table 1: the release versions of the different datasets used for the analysis in the manuscript should be mentioned.
4. The authors mention that the results presented in EV1 result from differences in gRNA efficiency. However, the two studies show opposite results, not significant results in one case and non-significant in the other. This could be better discussed.

5. In the introduction, the authors mention that genes "providing fitness" to a cell. Contributing to fitness could be a better term.

*. Comments related to code:

1. Description of the format of the output file is missing in the documentation of the python package.
2. Python package needs minor debugging to make it work right out of the box. e.g. requirements.txt contains double quotes which prevent the installation of the dependencies.
3. On the online tool, context analysis, "Choose gene for correlation:" only allows to scroll for genes starting with letter "A". Same for select query gene.
4. On plots showing the correlation of essentiality between pairs of genes, it would be useful to have interactive plots so one could easily find in which conditions a correlation is particularly weak for instance just by scrolling over the data point.

*. Comments related to figures:

1. Figure 2C: the variable represented by the size of point should be mentioned in the caption.
2. Figure 2D: the schematic represents 4 tandem repeats of SRE, however the text suggests that there are 8 of them.
3. Figure 3B: In order to determine the median change in essentiality score, as shown in colors, it may need a color legend.

*. Comments related to text:

1. In a few instances, Meyers et al, 2017 paper has been cited as a reference to the CCLE database. Authors could instead only use the citations of the CCLE database at those places.
2. In the 'Cell line selection tool' section of the 'Materials and Methods', it appears as if 0.15 is a version of R instead of Rtsne package, which I think should be the other way around.

Reviewer #2:

In this study, the authors presented CEN-tools to analyze the essentiality of the gene from two public genome-scale CRISPR screens. The CEN-tools could analyze the context-specific essentiality of genes and the gene co-association within a population of cells. In general, the manuscript writeup is clear. However, both functions provided are already available in the DepMap (<https://depmap.org/>). Please refer to "Enriched Lineages" and "Top Co-dependencies" with "Filter by" options in the DepMap portal. The slight difference present in CEN-tools does not warrant a new publication.

Reviewer #3:

This paper presents a new platform, CEN-tools, for exploring context-specific essentiality. It is based upon previously published datasets, in particular, large-scale CRISPR knockout screens from the Broad and Sanger institutes. These datasets can potentially provide extremely powerful novel insight, and it is good to have new tools for working with this data. In that regard, I think that CEN-tools could potentially be very useful. However, I find the examples of its application, which were chosen to demonstrate the power of the approach and comprise most of the paper, to be unclear,

difficult to follow, and in their current state, not enough to convince me of the utility of CEN-tools. I am not an expert in cancer/signaling, and it may be that these examples would make more sense if I were, but in that case, they should be more clearly explained for the general reader.

Major points:

- Can the authors comment on why they identified so many more core essential genes in the SANGER dataset than ADaM based on the same? It seems this study is likely just implementing a looser definition of essentiality. In that regard, it would be useful to include the properties of ADaM genes alone in Figs 1B and E.
- I found it very difficult to follow the SRF analysis. Fig 2C seems to be highlighting SRF as interesting for further analysis, although it doesn't particularly stand out from other genes. In Fig 2F, two transcription factors are tested, and an association is identified with SOX10. Why these two skin-specific TFs were chosen to be tested experimentally out of many possible? It's not obvious to me how CEN-tools guided this, and it feels a bit like it is just being used for post-hoc justification of the experiment. I could be wrong, and perhaps this would make more sense to someone more familiar with the underlying biology, but it needs to be made far more clear what is going on here for the general reader.
- I also found the analysis in Fig 3 difficult to follow. Again, it may make more sense to someone more familiar with the biology, but I'm not clear on the importance of the pathway enrichments in 3C, nor how the protein interaction network led to IGF1R being highlighted.
- As far as I can tell, none of the example analyses shown here could be replicated with the website, as only a single gene/tissue can be queried at a time. Thus it seems the utility of the website is severely limited, and it's hard to imagine many useful analyses being performed without using the Python package. The authors mention adding this functionality in the future in the Discussion, and I suggest they consider doing this as soon as possible.
- Why haven't the authors attempted to process the SANGER and BROAD data together? The explanation "to avoid variations due to institute origins of the cell lines" doesn't make much sense to me.
- The strategy for identification of essential core genes needs to be described in much more detail in the Methods.

Minor points:

- Silhouette score should be described in more detail - I don't think this is something most people are familiar with
- 1' and 2' therapeutic targets are not defined, except with reference to class of drugs from another paper
- Fig 1D - are these the only enriched protein complexes, or just a few that have been selected? The cutoff for significance should be indicated on the plot.
- Fig 1E - why not include the Sanger or Broad group?

Co-essentiality is clearly essential to many of the analyses that CEN-tools will be useful, e.g. Fig 2A. It would be very useful to include these co-essentiality associations directly within the platform.

We would like to thank the reviewers for their time and constructive comments, which we have addressed and as a result we are now submitting a much-improved manuscript and web server. Below we present a point-by-point response to their comments. For clarity, reviewer comments are coloured in gray, presented in italic fonts and annotated as C followed by the number of the reviewer and the rank number of each comment. Our respective responses are annotated with the letter R followed by the same numbers, in normal font, coloured black.

Reviewer #1:

The authors present a tool to navigate gene essentiality in different cellular contexts using large-scale CRISPR-Cas9 loss of function screens in human cell lines. Such large scale datasets are difficult to navigate for non-specialists but are a rich source of data to understand the relationships among genes. To fully exploit these resources, we need tools that allow to extract information among the importance of genes across cell lines, the similarity among genes, etc. Here, Sharma et al present such a tool that integrates information on some of the largest CRISPR screens that have been performed thus far.

Major comments:

C1.1. I believe integration webtools such as CEN-tools.com are very useful for a quick look up for one's preferred gene. People who are at ease with the analysis of such data may not need the web interface. I would therefore encourage the authors to make the web interface as user friendly as possible. For instance, if I go to Network Analysis, I am not sure what I am looking at in terms of network. It would be useful to have a description of what each page can do at the top, giving instructions to the user and for instance say: If you select a tissue type and X, CEN will show you a network that represents XX and YY. If the tool is not easy to use for non-bioinformaticians, it may be of limited use because bioinformaticians may not need it.

R1.1. We thank the reviewer for this comment and appreciate the feedback on our user interface. We aim to make it as intuitive and user-friendly as possible. To this end we initially consulted with the UX team (User Experience) at the EMBL-EBI. We then created a small task force comprising 6 people from computational and experimental backgrounds, including both experts and non-experts in CRISPR essentiality screens from the EBI, AstraZeneca and the Sanger institute. We finally presented this work in two virtual lectures with over 50 participants

each, one in an academic setting (Sven Furberg Seminar in Bioinformatics and Statistical Genomics) and the other in industry (AstraZeneca), and collected feedback for improving our web server. Overall we performed 5 rounds of receiving feedback and improving the web server. Examples of changes we have made include:

- In the co-essentiality/co-expression context analysis tabs we have included a list of co-expressed/co-essential genes at the pancancer level so that the user can directly start browsing the tool using relevant genes.
- In all plots we now allow the user to hover over the points to see to what exact cell line a point refers to and provide options to zoom into specific parts of the plots.
- We have changed the instructions in the gene selection boxes to 'Start typing to select gene' rather than 'Select gene' so that it is clear what the user needs to do and doesn't think they have to scroll for 15K genes to find their gene of interest.
- In the network analysis tab we have removed all advanced options that were causing confusion for the creation of the initial network. The user can then choose to activate advanced controls and we have also added additional controls to make the interface more flexible.
- We have also added a toggle button under the network that says 'Need help interpreting network?' and once selected it provides a detailed explanation of the different types of information displayed.
- We have improved the documentation and added additional information in all the '?' information points around the website. We also provide detailed instructions on how to reproduce the figures presented in the manuscript as examples of applications of CEN-tools under the Examples tab of the documentation and in the manuscript Structured Methods section.

We welcome any additional input and will continue to improve the website with feedback of users, which will no doubt continue to be provided after the tool has been published.

C1.2 As reported by the authors, they have identified a set of core-essential genes that are robustly essential (context-independent). These genes are contained in one of few clusters they identify based on the essentiality probability distribution (Cluster #1 in Figure EV10 panel C). The genes in this cluster distinctively show a high probability of being essential. However, this cluster only classifies ~3-5% of the genes. Among the rest of the genes (97-95%) classified in other clusters, I wonder if it would be possible to identify genes that are robustly non-essential

and those that commonly show context dependent essentiality. Such information, if it is possible to obtain, may add an additional utility to the CEN-tools method.

R1.2 We apologise for the lack of clarity. In our study the 4 clusters that we have identified correspond to 1: essential, 2: context-specific 3: rare context-specific 4: non-essential, based on the cell lines used in the studies analysed. Looking at the essentiality profiles of each gene of interest on our website shows how essential it is for how many cell lines, i.e. whether they are robustly essential, non essential or often or rarely context-specific. The lists of these genes are provided as Table EV1 and have now been annotated with these labels for clarity instead of cluster numbers.

C1.3 From the essentiality probability distributions (as shown in Figure EV10 panel C), it appears that a fraction of genes show robustly unimodal essentiality probability distributions (Cluster #3 and #4) with their peaks at low probability for being essential. Could such genes be classified as robustly non-essential?

R1.3 Cluster 4 does represent genes that are likely to be robustly non-essential, since they have very low probability of being essential across all the cell lines tested. Cluster 3 however, has a small peak at the region of the distribution where the probability of being essential is 1. As such, we classify these as 'rare context'-specific. To clarify this we have included annotations in Table EV1 for the different clusters.

C1.4 Overall, what fraction of the genes show intermediate probability of being essential, representing genes that commonly show a context dependent essentiality?

R1. 4 Clusters 2 and 3 comprise genes that fall in this category. Cluster 2 represents genes that are context-specific in several contexts, whereas Cluster 3 those that are context-specific in only a small number of cell lines (rare context-specific). We have now added the percentages of genes belonging in each cluster with clusters 2 & 3 representing the context and the rare context-specific genes respectively, in Appendix Figure S11.

Minor comments:

C1.5 1. It would be useful to include page numbers for peer review

R.1.5 We have now added page numbers to the revised manuscript.

C1.6 2. For the sake of stringency, in the integration of CEN with human interactome, authors excluded "dependent genes having median essentiality below than or equal to -1.5 in NRAS

mutant skin cell lines, edges having STRING interaction score below 400". In my opinion, the authors should explicitly mention the rationale for setting such thresholds.

R1.6 In the revised manuscript, we have changed the definition of 'Essentiality score' from scaled bayes factor to scaled log-fold changes, as this facilitates interpretation of these scores and it is what is also used in the Project Score for such analyses (personal communication). We have added a paragraph in the methods section to explain these calculations for all datasets (page 20, Step 3 under heading "**Data collection and curation**"):

Using this new definition of essentiality scores, we originally intended to define the cut-off of median essentiality score at 0.5, which would imply that at least 50% of cell lines in a given group had an essentiality score of at least 50% of that is expected of essential genes. However, we soon realised that this score alone tended to favour selecting associations for core essential genes, in which scores tend to be consistently high and disfavour associations for rare-contexts in which the sample numbers are low. This type of selection also omitted important edges if the dynamic range of the essentiality score was high; an example of this is shown below using *IRF4* in haematopoietic and lymphoid cell lines as a case study. Here, the median score of *IRF4* was 0.37, which precluded it from being selected as an essential gene of haematopoietic and lymphoid although it has significantly high essentiality (Figure 1 in this document). As *IRF4* is essential mainly in myeloid subtype within the haematopoietic and lymphoid tissue (Figure 1) the stratification within the tissue type led to high dynamic range of the essentiality scores and low median score.

Taking this into consideration, we devised a multiple thresholding system depending on how much dynamic range one would expect within the group being tested. While testing tissue alone, in which one expects a high level of confounders and therefore a much higher dynamic range of essentiality score distributions (e.g. different mutational backgrounds and cancer subtypes), we used a lenient threshold of a median essentiality score of 0.2 and a low confidence cutoff value 1 ($0.05 < p\text{-value} < 0.01$). For subtypes (which would still have stratifications according to mutational backgrounds, but a smaller dynamic range of essentiality score distributions), we increased the median essentiality score threshold to 0.3. When comparing a single mutational background within a particular tissue type or subtype, we increased the threshold to 0.4 and in some cases also further omitted the lowest-confidence (confidence cutoff 1) p-values.

Finally, should the users choose to use an alternate thresholding system according to their needs, the website allows easy adjustment of all the above mentioned thresholds.

We have now detailed this thresholding system and further recommendations on how to select appropriate thresholds in the methods section “**Applying advanced thresholds for network generation in CEN-tools**” in page 28.

Figure 1: Essentiality score distribution of *IRF4* in different tissue types (top panel) and in different cancer types within haematopoietic and lymphoid tissue lineage (bottom panel).

For the protein interactions, we used 400 mainly because this is the STRING default score on which our analysis is based on.

C1.7 3. Table 1: the release versions of the different datasets used for the analysis in the manuscript should be mentioned.

R1.7 We have included this information in the Reagents and Tools table wherever applicable.

C1.8 4. The authors mention that the results presented in EV1 result from differences in gRNA efficiency. However, the two studies show opposite results, not significant results in one case and non-significant in the other. This could be better discussed.

R1.8 It is indeed the case that the two studies show opposite profiles for some genes. The gRNA efficacy is determined by a number of factors including gRNA design and the screen duration. There exists differences between the projects in terms of both of these factors. The gRNA library used in BROAD is different from the one in SANGER and the number of days are also different (14 days for SANGER compared to 21 days for BROAD).

Sometimes the reasons for failing to identify essential genes could be because these genes are 'late essential genes' (genes that deplete after ~16 days in culture) but the cells are sampled at an earlier time point. However, this does not explain the situation we see here, as the BROAD dataset screen duration was longer than the SANGER screen duration. In the discussion section, we discuss how some of the discrepancy lies in genes that are very likely to be essential (such as components of histones and proteasome machinery, which are known to be 'early essential genes'). Therefore, we draw the conclusion that the low essentiality we observe for the BROAD dataset could in part be because of the gRNA efficacy. The third paragraph of the Discussion section is dedicated to these points and we have further added the following:

“In some cases it is known that 'late-essential' genes are not identified as essential genes if the screen endpoint is at an earlier time-point than the time it takes for the cells with the mutation in these genes to completely drop-out of the population. However, the fact that BROAD screens had a later endpoint than SANGER screens and that these genes are known to be 'early-essential' genes suggests that it is more likely that the which suggests that the gRNAs used for these particular genes in the BROAD study were of lower efficacy.”

C1.9 5. In the introduction, the authors mention that genes "providing fitness" to a cell. Contributing to fitness could be a better term.

R1.9 We have amended the term.

*. Comments related to code:

C1.10 1. Description of the format of the output file is missing in the documentation of the python package.

R1.10 We have added the description for the output format in the documentation of the package.

C1.11 2. Python package needs minor debugging to make it work right out of the box. e.g. requirements.txt contains double quotes which prevent the installation of the dependencies.

R1.11 We thank the reviewer for pointing out this. The "requirements.txt" file was corrected and the package has been further tested.

C1.12 3. On the online tool, context analysis, "Choose gene for correlation:" only allows to scroll for genes starting with letter "A". Same for select query gene.

R1.12 We agree that it is somewhat confusing. Unfortunately this is a feature of Rshiny, which was used to set up the entire web service and can not be changed to include all genes. Once a letter is typed then the list is updated to include all the genes with that letter. There is a clarification for this in the ? info button that is next to these fields. To avoid the confusion we have now changed the instructions to 'Start typing to select gene'.

C1.13 4. On plots showing the correlation of essentiality between pairs of genes, it would be useful to have interactive plots so one could easily find in which conditions a correlation is particularly weak for instance just by scrolling over the data point.

R1.13 This feature has now been added. We have also added the option to zoom into specific parts of the plot to facilitate selection of specific points in case these are very close in the plot.

*. Comments related to figures:

C1.14 1. Figure 2C: the variable represented by the size of point should be mentioned in the caption.

R1.14 To make the section on using CEN-tools to identify new gene-gene relationships clearer, we simplified Figure 2C to show only the network that was used in the section: There was no

need to show a scatterplot of all genes since we only considered the skin-specific transcription factors, and it was causing confusion. The legend has also been updated accordingly.

C1.15 2. Figure 2D: the schematic represents 4 tandem repeats of SRE, however the text suggests that there are 8 of them.

R1.15 Thank you for pointing this out. We have now found out that we wrongly assumed that it contained 8 tandem repeats based on another construct we had previously used. We used a commercial construct for this (Qiagen) and upon further investigation we have now found out that the number of repeats were different for different reporters and they have experimentally optimized numbers to maximize the sensitivity and specificity of each assay (<https://www.qiagen.com/gb/resources/resourcedetail?id=6fc8144b-010b-43d8-9125-bf541a5d9c9c&lang=en>). They have not revealed the exact number of tandem repeats for this specific construct. We have provided the catalog number of the exact construct we have used to enable replicability. In the manuscript, we have removed the exact number and replaced it with 'multiple tandem repeats'.

C1.16 3. Figure 3B: In order to determine the median change in essentiality score, as shown in colors, it may need a color legend.

R1.16 A color legend has been added to Figure 3B.

*. Comments related to text:

C1.17 1. In a few instances, Meyers et al, 2017 paper has been cited as a reference to the CCLE database. Authors could instead only use the citations of the CCLE database at those places.

R1.17 The citation used is that requested by the portal from which we extracted the dataset. Nevertheless we have now added the CCLE-specific citation as well.

C1.18 2. In the 'Cell line selection tool' section of the 'Materials and Methods', It appears as if 0.15 is a version of R instead of Rtsne package, which I think should be the other way around.

R1.18 Thank you for pointing this out. We adjusted the sentence such that now it should be clear that 0.15 is the version of the Rtsne package and 3.6.1 is the version of R used. This is also now detailed in the Reagents and Tools table.

Reviewer #2:

C2. In this study, the authors presented CEN-tools to analyze the essentiality of the gene from two public genome-scale CRISPR screens. The CEN-tools could analyze the context-specific essentiality of genes and the gene co-association within a population of cells. In general, the manuscript writeup is clear. However, both functions provided are already available in the DepMap (<https://depmap.org/>). Please refer to "Enriched Lineages" and "Top Co-dependencies" with "Filter by" options in the DepMap portal. The slight difference present in CEN-tools does not warrant a new publication.

R2 We thank the referee for their time to review our manuscript. We regret that they don't find it of sufficient novelty for publication and respectfully disagree. While CEN-tools indeed includes the analyses provided in the DepMap portal under the links mentioned, our tool includes several additional functionalities and visualisations. In addition, as the major novel component that significantly facilitates the interpretation of gene essentiality data at a systems level, we present the context-specific essentiality networks, or CENs. We showcase the use of CENs for discovering novel cancer type-specific gene essentialities in the paper as well as targets for specific cancer types, such as the NRAS-mutant melanoma. To be clear, we do not suggest that CEN-tools will function to replace DepMap, rather it provides a complementary, and we (and several users so far) believe a meaningful way to analyse these rich datasets and extract added value, while making them accessible also to non CRISPR-expert users.

Below we list the new analyses and components of CEN-tools, which are not features of the DepMap portal:

- The main and most important difference, is that we introduce Context-specific Essentiality Networks (CENs) which allow the navigation of essentialities in the selected contexts and provides a systems view for these allowing better interpretation. To illustrate this point: when looking at the top co-dependencies for SOX10 in DepMap, one can identify genes such as BRAF, DUSP4, MAPK1, and MITF. However, the co-dependencies of SOX10 with BRAF is a result of co-occurrence of expression of SOX10 in skin cell lines and enrichment of BRAF mutations in melanoma cells. SOX10 is essential in skin cell lines regardless of BRAF mutation (one can examine this in the context tab of CEN-tools--context analysis--mutation tab- also displayed in Figure 2A). Figure 2A and Appendix Figure S4 in the manuscript make a direct comparison of networks generated from correlated essentialities vs CEN-networks. We would hence argue that this is not merely a slight difference from visualisation but the

network representation of CEN-tools provides a more interpretable overview of the functions that are linked in a specific context.

- As the backbone for our service, we present a new analysis for identification of core essential genes and more than double the previously known ones. This leads to an improved identification of core essential, context-specific essential and non-essential genes in these cell lines. For individual genes, we have additionally generated patterns of essentiality probability, which are easily navigable by users through our website. This analysis has enabled us to pinpoint specific cases in which the essentiality profiles across the two projects have complete opposite patterns, or differ due to the differences in cell culture times. This could have implications on how the analysis resulting on the two different projects could lead to different conclusions. In the revised manuscript and webserver we also include the analysis of the Integrated dataset submitted recently to bioRxiv (Pacini et al. 2020).
- The DepMap portal, like CEN-tools, displays the datasets as correlations between two given types of data for specific genes or compounds. In CEN-tools, we provide additional representations for cross-tissue/context comparisons, including multi-group statistical tests e.g. selecting BRAF as a gene of interest will display the distribution of the essentiality scores in each tissue and will show that it is significantly more essential in the context of skin.
- In the revised version of the web server we have now included integration with STRING protein-protein interaction networks and functional enrichment of the resulting networks. This, as showcased in Figure 3, where we identify IGF1R as a potential target for NRAS-mutant melanoma, is very useful for identifying potential new targets for cancer sub-types with specific mutational or expression backgrounds.

The above points, along with positive feedback that we have received from users, strongly support that this new service will be a valuable addition to those provided by the DepMap portal.

Reviewer #3:

This paper presents a new platform, CEN-tools, for exploring context-specific essentiality. It is based upon previously published datasets, in particular, large-scale CRISPR knockout screens from the Broad and Sanger institutes. These datasets can potentially provide extremely powerful novel insight, and it is good to have new tools for working with this data. In that regard, I think that CEN-tools could potentially be very useful. However, I find the examples of its

application, which were chosen to demonstrate the power of the approach and comprise most of the paper, to be unclear, difficult to follow, and in their current state, not enough to convince me of the utility of CEN-tools. I am not an expert in cancer/signaling, and it may be that these examples would make more sense if I were, but in that case, they should be more clearly explained for the general reader.

We thank the reviewer for their time and comments. We apologise for the lack of clarity in the examples and we have attempted to clarify them in the text (please see specific responses below).

Major points:

C3.1 Can the authors comment on why they identified so many more core essential genes in the SANGER dataset than ADaM based on the same? It seems this study is likely just implementing a looser definition of essentiality. In that regard, it would be useful to include the properties of AdaM genes alone in Figs 1B and E.

R3.1 We thank the referee for the comment. The ADaM algorithm is based on “the membership threshold” in which the algorithm identifies the minimal number of cell lines from a given tissue in which the inactivation of a gene should exert a reduction of viability in order for that gene to be considered a core-fitness. In CEN-tools we do not consider the cell lines in terms of their tissue of origin in determining core-essentiality but rather use a probability distribution-based clustering for all available cell-lines after logistic regression to determine if a gene is core-essential. Our ability to detect a larger number of core essential genes is most likely due to us not restraining the analysis to tissue membership. However, based on the performance of our regression algorithm (Appendix Figure S11) and the probability distributions of the genes that we have identified as essential, along with the fact that we identified these through two analyses of separate datasets, we are confident that our list of core-essential genes is robust. We have included the requested panels for ADaM genes in Fig 1E. For Figure 1B we would rather not include it in the paper as a main Figure as this is not an evaluation of ADaM, which has previously been published, and it dilutes the message. For your information the plot is pasted below. If you would still like us to include it in the main figure we can do this.

C3.2 I found it very difficult to follow the SRF analysis. Fig 2C seems to be highlighting SRF as interesting for further analysis, although it doesn't particularly stand out from other genes. In Fig 2F, two transcription factors are tested, and an association is identified with SOX10. Why these two skin-specific TFs were chosen to be tested experimentally out of many possible? It's not obvious to me how CEN-tools guided this, and it feels a bit like it is just being used for post-hoc justification of the experiment. I could be wrong, and perhaps this would make more sense to someone more familiar with the underlying biology, but it needs to be made far more clear what is going on here for the general reader.

R3.2 We apologise for the lack of clarity. Using CEN-tools, we identified that *SRF* was among one of the TFs that had higher essentiality in skin compared to other tissues. The reason that *SRF* was interesting was the following: Over half of the skin cell lines we have in the dataset are *BRAF*-mutant, meaning that they have the MAPK pathway (downstream of which is the *SRF* transcription factor) constitutively active. Therefore one could assume/hypothesize that *SRF* is essential because *BRAF* is mutated and the cells are addicted to the MAPK pathway. However, even though *SRF* was indeed essential in skin, we surprisingly found that there was no statistical association between the mutational status of *BRAF* and *SRF* essentiality. We therefore sought for other candidates that could be creating the 'context' for this essentiality in skin for *SRF*. The skin-specific transcription factors were the obvious first choice since tissue-specific TFs are often the ones driving the tissue-specific differences in cell molecular profiles. We tested *MITF* and *SOX10* since they were the two most significantly essential TFs in skin and

we found the association with *SOX10* and not *MITF*. To be clear, we do not claim that *SOX10* is the only gene that has a relationship with *SRF*. However, due to COVID-19-related lab closures we were unable to test further genes, but in any case felt that it is beyond the scope of this study to identify all the genes associated with *SRF* essentiality. We included this example to showcase one functionality of CEN-tools. Specifically the case study with CEN-tools here shows that we can test a hypothesis about gene-gene essentiality associations (in our case that *SRF* is essential because of the *BRAF* mutation), invalidate it (there was no association), identify candidates to test for finding new gene associations (we picked the top two most significant skin-essential TFs identified by CEN-tools).

We have re-written that section and changed Figure 2C to make all this clearer.

C3.3 I also found the analysis in Fig 3 difficult to follow. Again, it may make more sense to someone more familiar with the biology, but I'm not clear on the importance of the pathway enrichments in 3C, nor how the protein interaction network led to IGF1R being highlighted.

R3.3 We apologise for not making it clearer in the text: The importance of the pathway enrichment is that they highlight potential target pathways for *NRAS*-mutant melanoma, which currently has limited therapeutic options. *IGF1R* was specifically highlighted because a) it is closest to *NRAS* in the network and therefore more likely to be a true association b) there are many effective drugs targeting it, and c) it is already targeted in *BRAF*-mutant melanoma when it is resistant to MEK and BRAF inhibitors. As such it would be an excellent target to evaluate in the context of *NRAS* melanoma.

We have now added the sentence :”These represent potential target pathways for *NRAS*-mutant melanoma.” after we mention the enriched pathways. We also added “and to highlight potential candidate target genes and pathways (Lord et al, eLife 2020)” as a justification for the data integration. In the cited paper they show that genes whose protein products interact are more likely to have genetic interactions. Therefore, in our case, these could be ideal targets in the context of specific mutations. We also added after mentioning *IGF1R*, the explanation “for which several drugs already exist, but are not used in the context of *NRAS*-mutant melanoma to the best of our knowledge”.

C3.4 As far as I can tell, none of the example analyses shown here could be replicated with the website, as only a single gene/tissue can be queried at a time. Thus it seems the utility of the website is severely limited, and its hard to imagine many useful analyses being performed

without using the Python package. The authors mention adding this functionality in the future in the Discussion, and I suggest they consider doing this as soon as possible.

R3.4 All the analyses were done using the website except for overlaying the protein-protein interactions on the essentiality network. We have now included in the documentation and the Structured Methods of this manuscript step by step manuals for reproducing these examples, as well as improved documentation for other functionalities. We have also made significant effort in improving the user-friendliness of the website so that the availability and use of its different functionalities is more intuitive (please see response R1.1 for details on the things we modified). In addition, in this revision, we have implemented the integration with the STRING database protein-protein interaction networks in the web server and therefore all figures are now completely reproducible using only the web server. We have changed the figures in the manuscript to reflect the output figures that are directly generated from the website.

C3.5 Why haven't the authors attempted to process the SANGER and BROAD data together? The explanation "to avoid variations due to institute origins of the cell lines" doesn't make much sense to me.

R3.5 We thank the reviewer for the comment and they are right that an integrated analysis would provide more power to the web server. As it happens, the integrated analysis of the SANGER and BROAD datasets was just submitted to bioRxiv recently (Pacini et al. 2020), and we have now updated the web server and Appendix Figure S10 to include this.

C3.6 The strategy for identification of essential core genes needs to be described in much more detail in the Methods.

R3.6 We have now restructured the methods to read like a protocol, and included additional details to enhance understanding and reproducibility of this pipeline. The code used is also provided in our gitlab page.

Minor points:

C3.7 Silhouette score should be described in more detail - I don't think this is something most people are familiar with

R3.7 We have now added more detailed clarifications for these in our explanation of the core essential gene identification in the Results and Methods sections. In the latter specifically we added the sentence in the "**Identification of core essential genes**" section (point 3).

“The silhouette method compares the distance of each point to points in the same cluster with the distance to points in the neighboring clusters. A high silhouette score means that the points are well placed within a cluster and the clusters are well separated, allowing a robust definition of the number of clusters and the assignment of samples.”

C3.8 1' and 2' therapeutic targets are not defined, except with reference to class of drugs from another paper

R3.8 These referred to therapeutically tractable targets that are also associated to biomarkers vs not as defined by the Behan et al 2019 paper. However, as separating the therapeutically tractable targets to two categories didn't add anything to our message, for simplicity we updated the Figure 1C to group them and updated the legend to include an explanation of what these refer to.

C3.9 Fig 1D - are these the only enriched protein complexes, or just a few that have been selected? The cutoff for significance should be indicated on the plot.

R3.9 It included 10 representative complexes from those enriched. For simplicity we have now modified figure 1D to include the top 15 most enriched complexes. All the displayed complexes in the modified figure 1D have an adjusted p-value<0.01 and the cutoff significance is displayed as a red line in the figure and described in the legend.

C3.10 Fig 1E - why not include the Sanger or Broad group?

R3.10 According to the literature the expression values of the commonly essential genes should be high. In this figure, we were interested in comparing the basal expressions of the core genes (both known and newly predicted) inside the normal tissue. Since we were only considering as core genes those predicted from both the BROAD and SANGER datasets, if they were identified in only one of the projects these were not included. Appendix Figure 10 includes these results for the integrated dataset that was recently published in bioRxiv (Pancini et al, 2020)

C.3.11 Co-essentiality is clearly essential to many of the analyses that CEN-tools will be useful, e.g. Fig 2A. It would be very useful to include these co-essentiality associations directly within the platform.

R3.11 There are several tools that provide the co-essentiality profiles across all the cell lines (e.g. PICKLE database). The main novelty of our tool is that it provides the CENs. These represent indeed the co-essentiality relationships but in a unique way, where the user can navigate the information on the context of these relationships. For example standard co-

essentiality associations would make a link between BRAF and SOX10. However, as we demonstrate in Figure 2A and Appendix Figure S4, our CENs clearly show that the link between these two is actually a confounder effect because of the fact that SOX10 is only expressed in skin and BRAF is very often mutated in skin. Therefore, we are providing a much more informative view of these co-essentialities and would rather not display them in the format that several other tools already do. For information purposes, we have now included the top 10 globally co-essential genes for any gene selected in the respective context-analysis tabs, to aid browsing of gene-gene relationships.

Thank you for sending us your revised manuscript. We have now heard back from the two reviewers who were asked to evaluate your revised study. As you will see below, they are both satisfied with the modifications made and they are supportive of publication. As such, I am glad to inform you that your manuscript is now suitable for publication, pending some minor editorial issues listed below.

We would ask you to address the following in a minor revision.

REFEREE REPORTS

Reviewer #1:

The authors have modified the tool and manuscript in a satisfactory manner.

Reviewer #3:

I am happy that the reviewers have not adequately addressed my original comments, and I think the paper is now suitable for publication

Corresponding Author Name: Sumana Sharma and Evangelia Petsalaki

Manuscript Number: MSB-2020-9698